# CODIFIED FINITE-STATE MACHINES FOR ROLE-PLAYING

**Letian Peng, Yupeng Hou, Kun Zhou, Jingbo Shang**
`{lepeng, yphou, kuzhou, jshang}@ucsd.edu`
University of California, San Diego

## ABSTRACT

Modeling latent character states is crucial for consistent and engaging role-playing (RP) with large language models (LLMs). Yet, existing prompting-based approaches mainly capture surface actions, often failing to track the latent states that drive interaction. We revisit finite-state machines (FSMs), long used in game design to model state transitions. While effective in small, well-specified state spaces, traditional hand-crafted, rule-based FSMs struggle to adapt to the open-ended semantic space of RP. To address this, we introduce Codified Finite-State Machines (CFSMs), a framework that automatically codifies textual character profiles into FSMs using LLM-based coding. CFSMs extract key states and transitions directly from the profile, producing interpretable structures that enforce character consistency. To further capture uncertainty and variability, we extend CFSMs into Codified Probabilistic Finite-State Machines (CPFSMs), where transitions are modeled as probability distributions over states. Through both synthetic evaluations and real-world RP scenarios in established artifacts, we demonstrate that CFSM and CPFSM outperform generally applied baselines, verifying effectiveness not only in structured tasks but also in open-ended stochastic state exploration.[1]

## 1 INTRODUCTION

In role-playing (RP) (Shao et al., 2023a; Wang et al., 2024b; Ran et al., 2025), character states are as vital as plot progression, shaping how a character reacts, evolves, and engages in unfolding narratives. Consider the "Mario" case in Figure 1: grabbing a "Super Mushroom" promotes him to "Super Mario" with new affordances (e.g., breaking blocks); a "Fire Flower" induces another state with ranged attacks. Effective RP likewise demands precise and expressive modeling of such state changes in open-ended, language-based worlds (e.g., when writing scripts for *"The Super Mario Bros. Movie"*).

Despite recent advances, LLM-based (Achiam et al., 2023; Dubey et al., 2024) RP agents struggle with state stability and behavioral coherence. Prompt-only transition heuristics often miss which experiential details are decision-critical, and they lack a transparent mechanism to carry state forward over time. Finite-state machines (FSMs), long used in game design, directly address these issues by specifying explicit states and interpretable transition rules, yielding deterministic, debuggable updates that preserve narrative consistency. However, traditional FSMs rely on hard-coded rules that do not scale to the fluidity and ambiguity of text. This paper tends to preserve the strengths of FSMs while relaxing the rigidity: we disentangle constraints into executable checks and leave space for exploration via semantic questions. It is feasible because the rationality of each constraint can be estimated from the profile and context, allowing precise rules where needed and flexible guards elsewhere (Peng & Shang, 2025).

We propose **Codified Finite-State Machines (CFSMs)**: using LLM-based extraction and coding (Xu et al., 2024; Jiang et al., 2024), we first prompt LLMs to extract key states involved in transition from the profile. Based on extracted states, LLMs then generate an executable transition function `get_next_state(state, scene, action)` that queries the scene through pre-defined helper calls `binary_question(text, question)`. CFSMs thus ground character logic in structured, interpretable transitions while remaining adaptable to open-ended semantics.

---

[1] Code: https://github.com/KomeijiForce/Codified_Finite_State_Machine

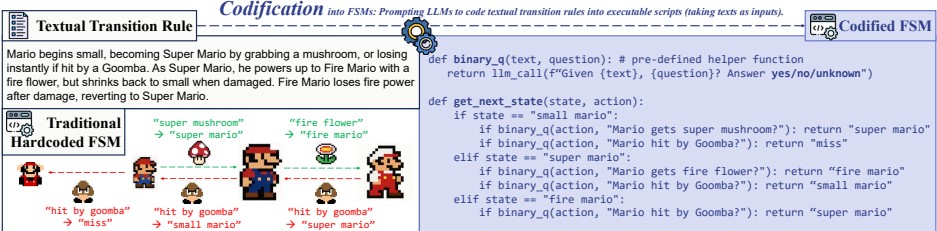

Figure 1: The Mario case to introduce finite-state machines into the role-playing field.

To capture more nuanced dynamics, we introduce **Codified Probabilistic Finite-State Machines (CPFSMs)**, which represent states as probability distributions, enabling multinomial-distribution and explainable transitions when multiple next states are plausible. Compared with prompt-only methods, CFSM and CPFSM reduce confusion and inconsistency, producing stable and believable role-play.

To implement CFSMs, we parse profiles to identify distinct states, including catch-all "unactivated" and "other" states, and then codify transition rules as executable logic conditioned on scene events. During role-play, each action is processed through this logic to update the current state, grounding subsequent outputs in a transparent trajectory. With logits available from the condition checker, CPFSM extends this framework into a probabilistic transition matrix that continuously updates state distributions. This mechanism captures subtle shifts in internal dynamics and supports multinomial-distribution reactions with explicit likelihoods, thereby increasing expressiveness and realism in open-ended RP.

Synthetic validation experiments, such as Mario's power-up transitions and stealth combat logic in Call of Duty, reveal that LLMs prompted directly with multi-action contexts often confuse states, misapply conditions, or hallucinate transitions. These findings reinforce that prompt-only RP models lack stable state modeling, while CFSM's explicit scaffolding significantly improves coherence.

We extend evaluation to real-world narrative tasks using the established plots in Fandom Benchmark (Peng & Shang, 2025), which includes over 5,000 role-play scenes across 83 characters, and find that CFSM and CPFSM improve behavioral consistency, transition traceability, and alignment with character-defined profiles relative to prompting and state-modeling baselines. CPFSM further supports the claim that probabilistic modeling yields more nuanced, explainable, and multiple potential behaviors, especially when multiple actions are plausible, validating codified FSMs, deterministic or probabilistic, as strong foundations for state modeling in LLM-driven role play. To clarify CFSM's practical contributions, we analyze three aspects: ablation shows explicit state registration is critical, with removal degrading action consistency and with transitions grounded in disjoint profile facets such as emotional stance and social role producing the strongest coherence; cost analysis introduces an $O(n+k)$ codification strategy that assigns default conditions to all $n$ states and overwrites only $k$ profile-defined transitions, enabling scalable construction compared with $O(n^2)$; and a case study illustrates faithful tracking of dynamic states over episode progress. Our contributions are three-fold:

- We propose Codified Finite-State Machines (CFSM), a framework that uses LLM-based extraction and coding to define interpretable character state transitions from textual profiles.
- We introduce Codified Probabilistic Finite-State Machines (CPFSM), enabling probabilistic state dynamics that support nuanced, multiple potential reactions in narrative contexts.
- We provide both synthetic and real-world evaluations, along with ablation and cost analysis, demonstrating that our codified approaches improve coherence, interpretability, and efficiency.

## 2 RELATED WORKS

**Role-playing.** Role playing (RP) tasks an agent with enacting a persona (e.g., goals, backstory, and constraints), while producing consistent dialogue or actions (Riedl & Bulitko, 2012; Riedl & Young, 2005). LLMs have dramatically expanded what RP agents can do (Shao et al., 2023b; Chen et al.), and an LLM-based agent augmented with memory (using relevance and recency heuristics) can simulate believable characters (Moore Wang et al., 2024; Yan et al., 2023). However, persistent personas remain hard: systems often drift in memory and behavior as narratives grow complex. Fine-tuning LLMs on character-specific or experience data (e.g., CharacterGLM (Shao et al., 2023b) and Neeko (Yu et al., 2024)), can improve identity consistency over prompting, but requires curated

data and is costly to scale up. In parallel, systematic mechanisms within the agent are being explored to guide behavior over long interactions, including reasoning-based action selection (Tang et al., 2025), structured memory (Cheng et al., 2025), and codified constraints (Peng & Shang, 2025).

**State Modeling.** State modeling makes latent variables explicit, together with rules to govern an agent and its world evolves over time (Corbett et al., 2000; Chomsky & Miller, 1958). It typically enumerates discrete states (e.g., idle and alert) and specifies condition- or event-triggered transitions (Joseph et al., 2013; Svete & Cotterell, 2023), yielding an interpretable control interface for RP. Histories can be summarized and organized hierarchically to maintain concise, decision-ready state descriptions. Recent efforts improve tracking and structure to stabilize transitions and avoid continuity errors, including graph-based memory (Li et al., 2024) and identity-driven hierarchical architectures (Sun et al., 2024). Codified Profiles (Peng & Shang, 2025) further compile textual descriptions into executable condition-checking logic, enabling persistent and controllably stochastic behavior. Yet transparent, conflict-resolving transition logic remains scarce in open-ended RP.

**LLM for Coding.** LLMs are widely used to synthesize, edit, and orchestrate executable code from language or reasoning specifications (Luo et al.; Li et al., 2022). Building on this capability, agentic and neurosymbolic paradigms offload parts of reasoning to code execution, markedly improving accuracy in reasoning (Yao et al., 2023; Gao et al., 2023) and embodied AI tasks (Wu et al., 2023; Liang et al., 2022). These directions motivate our use of an LLM to generate FSM logic: we leverage its knowledge and capability to write code that tracks states and triggers actions, ensuring that high-level plans (state transitions) adhere to a predefined logical schema. Recent work shows LLMs can also author or revise state machines directly: they modify FSMs (as code) from natural-language instructions in robotics (Yoneda et al., 2024; Gan et al., 2024) and multi-hop reasoning (Wang et al., 2024a; 2025b), demonstrating that nontrivial transition logic can be produced in code form.

## 3 CODIFIED FINITE-STATE MACHINE

### 3.1 PRELIMINARY AND DENOTATION

**Basic Role-playing System.** For a character $x$, the most basic role-playing system consists of a text generator (large language model in our discussion) $\mathbf{LLM}(\cdot)$ that generates a character response $r$ to a given scene $s$. The generation is guided by two types of instructions: a global instruction $I_g$, which encodes universal role-playing behavior, and a character-specific instruction $I_x$, which captures the identity, traits, or constraints of character $x$. The model samples the character's response by conditioning on all three components: $r = \mathbf{LLM}(s \mid I_g, I_x)$. The character state information discussed later is included in $I_x$ for grounding.

**State Modeling in Role-playing.** In role-playing systems, a scene $s$ can be decomposed into a sequence of observable actions $[a^{(1)}, a^{(2)}, \cdots, a^{(t)}]$, where each action $a$ may originate from character behavior or external environmental events. Alongside this explicit action sequence, there exists a parallel sequence of latent character states $[h^{(1)}, h^{(2)}, \cdots, h^{(t)}]$, where each $h$ represents the evolving internal state of a character, which captures psychological factors such as emotions and intentions. To capture it, role-playing systems often model state transitions using a Markovian formulation: $h^{(t+1)} = T(h^{(t)}, a^{(t)}, I_g, I_x)$. Here, $T(\cdot)$ denotes the state transition function, which updates the internal state based on the previous state $h^{(t)}$, the current action $a^{(t)}$, and both the global instruction $I_g$ and character-specific instruction $I_x$ that might contain certain transition rules. This structure is particularly designed under the limited context window of LLMs, allowing systems to track and compress character dynamics explicitly. A straightforward implementation of $T(\cdot)$ is to prompt the LLM to generate the next state $h^{(t+1)}$ conditioned on the tuple $(h^{(t)}, a^{(t)}, I_g, I_x)$ **(PromptTrans)**, enabling the model to simulate coherent and evolving character behavior over time. During the response, $h^{(t)}$ is appended to the prompt inside $I_x$ for grounding.

**Finite-State Machine.** A finite-state machine (FSM) is a formal structure for modeling stateful behavior, defined by the tuple $\mathcal{M} = (H, A, T, h^{(0)})$ where $H$ is the set of possible states, $A$ is the set of actions or events, $T : H \times A \to H$ is the state transition function, and $h^{(0)} \in H$ is the initial state. Unlike direct prompting of an LLM to output $h$ in the infinite set of natural language, FSMs

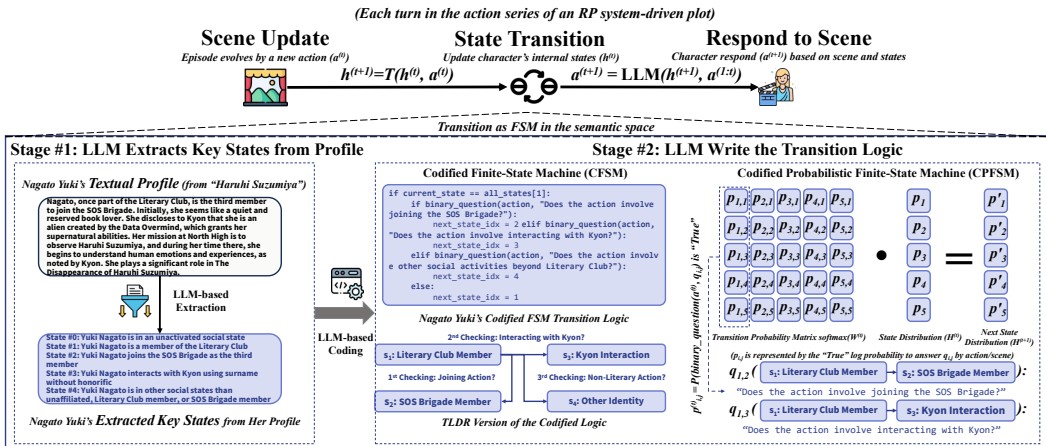

Figure 2: The frameworks of CFSM and CPFSM. More implementation-focused flow: Figure 11.

only maintain the transition between a set of key states. FSMs also provide an explicit and persistent representation of character states and their transitions. This explicit structure enables RP systems to (1) preserve long-term continuity beyond the LLM's context window, (2) enforce logical consistency of state transitions, and (3) integrate both parametric knowledge (model weights) and in-context knowledge (scene-specific updates) into an explicit and explainable evolution of character behavior. In this way, FSMs serve as a bridge between traditional symbolic state tracking and the open-ended semantic reasoning afforded by LLMs.

**Illustrative Example.** For clarity, consider *Jotaro Kujo* from *JoJo's Bizarre Adventure*. The global instruction $I_g$ enforces general RP coherence, while the character instruction $I_x$ encodes Jotaro's stoicism, terse speech, and *"Yare yare daze."* In a scene $s$ where an enemy Stand user appears, the observable actions may be $a^{(1)} = $ *enemy approaches* and $a^{(2)} = $ *enemy threatens*. An FSM uses a small set of discrete states (*neutral*, *alert*, *combat*, *post-battle*) with explicit transitions such as $alert \xrightarrow{enemy\ threatens} combat$. Taken the mentioned actions as inputs, FSM transitions Jotaro from $h^{(1)} = alert$ to $h^{(2)} = combat\text{-}ready$ via $h^{(t+1)} = T(h^{(t)}, a^{(t)}, I_g, I_x)$, guiding increasingly tense responses. During generation, the LLM is grounded in the current state, yielding character-faithful lines like *"Yare yare daze... Guess I'll have to deal with you."* This illustrates how structured state tracking supports coherent and identity-consistent RP behavior.

## 3.2 CODIFIED FSMs

One character can have multiple FSMs modeling different behavioral dimensions (e.g., emotion and stance). For simplicity, we illustrate the codification process of a FSM instance $\mathcal{M} = (H, A, T, h^{(0)})$, where each component is constructed via LLM-based codification from textual profiles and scenes.

**Key State Extraction.** The first step in constructing a CFSM is to extract a discrete set of key states $H = \{h_1, h_2, \ldots, h_n\}$ from a character profile $P_x$. These states capture semantically meaningful internal conditions (e.g., angry, curious) that guide role-play. Rather than manually defining all possible states, CFSM leverages LLMs to generate $H$ via $H = \text{LLM}(P_x \mid I_{\text{extract}})$ where $I_{\text{extract}}$ is an instruction prompt for state enumeration. This allows character logic to be grounded in interpretable and controllable state spaces while remaining flexible to profile formats. To cover all possible states of the character, $H$ reserves $h_1$ for "Unactivated" (also for initialization as $h^{(0)}$) and $h_n$ for "Other".

**Codification.** Given the extracted states $H$ and profile $P_x$, the next step is to codify the state transition function $T : H \times A \to H$. Rather than hard-coding rules, CFSM uses the LLM to generate a function by prompting it to map combinations of state-action pairs to the next state, grounded in the character's behavioral logic from $P_x$: $T(h, a) = \text{LLM}(h, P_x \mid I_{\text{codify}})$. It captures how the character would typically evolve in reaction to different stimuli, embedding both semantic and symbolic reasoning in the transition logic. Such semantic reasoning is supported by a `binary_question(text, question)` function embedded inside the code, which is executed for condition checking based on question $q$ (pre-defined during codification) by the RP LLM or a specifically trained discriminator.

**Mechanism.** Once codified, the CFSM operates by simulating character state transitions as a function of current state and observed action $h^{(t+1)} = T(h^{(t)}, a^{(t)})$ (`get_next_state(state, scene, action)`). This lightweight mechanism enables external systems to simulate the character's internal state evolution across a sequence of events without prompting the LLM per step. By separating codification from execution, CFSMs can be further cached, reused, audited by humans.

### 3.3 CODIFIED PROBABILISTIC FSMS

While CFSMs model state transitions deterministically, real-world RP often involves ambiguous signals and stochastic behaviors. To capture this uncertainty, we extend CFSM to Codified Probabilistic Finite-State Machine (CPFSM). It replaces discrete state transitions with a probabilistic state distribution, enabling more expressive and nuanced modeling of character dynamics.

**Probabilistic State Representation.** Instead of a single active state $h_i \in H$, CPFSMs maintain a state probability distribution: $P_i^{(t)} = [p_1^{(t)}, p_2^{(t)}, \ldots, p_n^{(t)}] \in \Delta^{n-1}$, where $p_k^{(t)}$ represents the likelihood that the character is in state $h_k$ at time step $t$, and $\Delta^{n-1}$ is the probability simplex over $n$ possible states. This distribution evolves over time as new actions are observed.

**Transition Matrix.** To update the state distribution, CPFSMs employ a logit-weighted transition matrix $W^{(T)} \in \mathbb{R}^{n \times n}$, where each element $w_{i,j}^{(t)}$ represents the logit score of transitioning from state $h_i$ to $h_j$ given the current action $a_i$. These logits are derived by prompting the LLM with pre-defined transition questions $q_{i,j}$ (derived from the profile) and the current action context, and $w_{i,j}^{(t)}$ is then normalized via the softmax function to transition probability $p_{i,j}^{(t)}$. The final transition is $P^{(i+1)} = \text{softmax}(W^{(t)}) \cdot P^{(i)}$, where $w_{i,j}^{(t)} = \texttt{binary\_question}(a^{(t)}, q_{i,j})$, represented by the log probability of "True" prediction from the classifier.

**Inference and Grounding.** Although the internal representation is probabilistic, CPFSMs maintain compatibility with deterministic RP systems by grounding the response at each time step in the most probable state $h_k^{(t)}$ where $h_k = \arg\max_i p_i^{(t)}$. This allows CPFSMs to guide role-playing outputs while preserving interpretability and controllability. Furthermore, because transitions are conditioned on both structured prompts and soft distributions, CPFSMs support fine-grained adaptation, uncertainty modeling, and richer behavioral diversity compared to their deterministic counterparts.

### 3.4 IMPLEMENTATION OVERVIEW

Given a new action, we first check its relevance to the character, which prevents updates based on unrelated or absent events. If relevant, we then classify whether the character is the agent (active) or recipient (passive) of the action to select the corresponding codified transition logic (both FSMs codified in advance). More details are in the prompt and pipeline (Appendix C and F) implementation.

Using the character's current state and the new action, the system runs the codified transition function (deterministic in CFSM, probabilistic in CPFSM) to compute the next internal state, where `binary_question` is supported by a smaller and faster discriminative model such as `gpt-4.1-mini` or a distilled classifier to evaluate semantic conditions efficiently, while the main response generation is handled by a larger RP model like `gpt-4.1`[2].

| Aspect | CFSM | CPFSM |
|---|---|---|
| Transition Type | Deterministic | Probabilistic |
| Granularity | Coarse-grained | Fine-grained |
| Speed | Faster | Slower |
| Logit | Not required | Requires |
| Transition Bias | Greedy | Distributional |
| Branching Logic | Multi-path checks | Single-step |
| State Distribution | Not available | Available |

Table 1: CFSM/CPFSM Comparison.

Finally, after computing the character's new internal state, the RP model generates a response grounded in that state and the current scene context. This structured flow ensures that state transitions are interpretable, profile-consistent, and dynamically updated over the narrative.

---

[2]We don't experiment on `gpt-5` because it does not support $T = 0.0$ at the time of submission.

**CFSM vs. CPFSM**   Table 1 summarizes the trade-offs between CFSM and CPFSM. CFSM offers faster inference and greater compatibility with black-box LLMs by avoiding logit computation, making it suitable for interpretable and flexible branching logic.  In contrast, CPFSM provides finer-grained, less biased transitions by modeling state uncertainty as distributions, making it ideal for nuanced or stochastic role-playing scenarios.

## 4   SYNTHETIC VALIDATION

We first validate the relative limitation (to FSMs) of LLMs in maintaining the character's states in role-playing. To do so, we select three classic and well-known state transition models, which are used to generate state transition sequences to reveal the limitations of LLMs in modeling character states. These state transition models include Mario State, Call of Duty Enemy, and Tyrion's Westeros Travel.

- **Mario State (Item Interaction).** Mario's state transition in our experiments represents a simplified version of the game, where Mario has "small", "super", "fire", and "miss" states. Mario transitions between states based on actions "get a super mushroom", "get a fire flower", "hit by a goomba".
- **Call of Duty Enemy (Reaction Interaction).**  The enemy state machine models typical FPS opponent behavior with states "idle", "alert", "engaged", "retreat", and "death". Transitions are triggered by player actions such as making noise, revealing position, sustained fire, or direct elimination, capturing the dynamics of stealth, combat, and survival.
- **Tyrion's Westeros Travel (Map Interaction).**  Tyrion's movement is modeled as transitions across regions of Westeros: "the westerlands", "the riverlands", "the vale", "the crownlands", "the stormlands", "the reach", "dorne", "the north", and "the iron islands". Each transition corresponds to a directional travel action (e.g., north, northeast), enforcing geographical adjacency on the map.

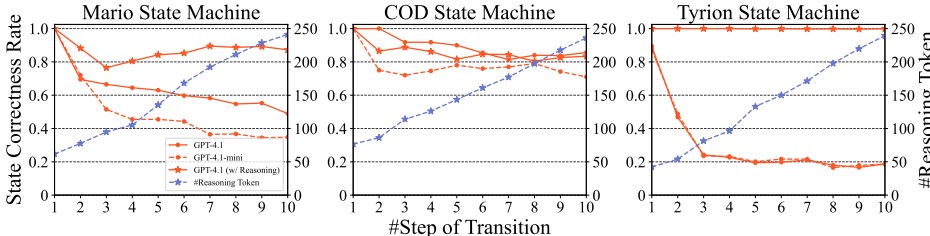

Figure 3: Synthetic validation of LLMs' limitation in state transition understanding.

We place the FSM for trace generation in the Appendix C. For each FSM, we generate 10,000 transition paths under a random action selection policy. To ensure balanced evaluation, we further sample 100 paths terminating in each possible state (e.g., for the 4 Mario states, this yields $4 \times 100 = 400$ test paths). The evaluated LLM is prompted with the textual transition rules, an initial state, and a sequence of actions, and it must predict the correct final state from the full state set. We repeat this procedure for path lengths from 1 to 10, gradually increasing task difficulty.

Figure 3 reports the results of these synthetic validation experiments, comparing `gpt-4.1` and `gpt-4.1-mini`, with and without chain-of-thought prompting. **CFSM results are omitted, as it consistently achieves** $100\%$ **accuracy with forward time proportional only to the number of transitions.** The LLMs, in contrast, show a clear decline in accuracy as path length increases, particularly without chain-of-thought reasoning.  While chain-of-thought substantially improves accuracy by effectively simulating the FSM, it incurs a significant efficiency cost, requiring roughly $25\times$ the forward time of CFSM. Case studies in the Appendix J illustrate that chain-of-thought predictions actually mimic explicit FSM simulation, underscoring the value of CFSM in delivering both stable transition and high computational efficiency.

## 5   REAL PLOT EXPERIMENT

### 5.1   EXPERIMENT SETUP

**Fandom Benchmark**   (Peng & Shang, 2025) is constructed from character profiles and structured story summaries sourced from Fandom, focusing on behavior-centric role-playing evaluation. For each narrative segment, scenes are paired with extracted character actions and guiding questions

| Artifact | Scene | QA Pair |
|---|---|---|
| Haruhi Suzumiya | Haruhi drags the SOS Brigade around to do various summer vacation activities, including swimming in a pool, going to an O-bon festival, playing with fireworks, and bug hunting for the 15,499th time. Kyon is plagued by an increasing sense of déjà vu as the activities continue. Again, Asahina calls Kyon in a panicked state, asking him to meet her. | **Question:** What action does Kyon take after receiving a panicked call from Asahina during the endless summer loop?
**Reference:** Kyon meets up with her, Nagato, and Koizumi, who again explain that the world has been experiencing an endless time loop from August 17 to August 31 because Haruhi feels she still has things she must do during summer vacation. |
| A Game of Thrones | ..., Jon Snow works at his sword practice, angry that Catelyn thought it would be inappropriate that a bastard should attend. His uncle Benjen Stark, First Ranger of the Night's Watch, arrives to join the feast, and Jon asks him to take him back to the Wall with him. Benjen agrees to consider it. | **Question:** How does Tyrion Lannister address Jon Snow's sensitivity about his illegitimacy during their conversation?
**Reference:** Tyrion Lannister then arrives and talks to Jon, suggesting that he is too pricklish and quick to take offense when his illegitimacy is pointed out. |

Table 2: Examples of data samples used in the Fandom Benchmark.

that constrain responses without revealing the answer. During evaluation, the role-playing LLM receives a spoiler-free profile, the scene, and the question, and must predict the character's next action. The benchmark spans six major artifacts (*Haruhi*, *K-On!*, *JOJO*, *FMA*, *AGOT*, *ATLA*), covering 83 characters and 5,141 scenes across diverse media genres. Its scene-action pairs are arranged in a temporal sequence, which enables state updating experiments along with the episode evolution. Background context of characters and episodes involved in these artifacts, together with running examples can be found in Appendix M. Table 2 showcases two data samples from the benchmark for easier understanding of the benchmarking procedure.

**Natural Language Inference (NLI) Score and Best@K Responses.** Predictions are scored with an LLM-based (`gpt-4.1`) reference-prediction NLI following the prompt setup in the original benchmark, assigning 100 for entailment, 50 for neutral, and 0 for contradiction. The NLI accuracy is also validated to be consistent with human evaluation in previous work. We further manually validate 200 cases predicted as "entailed", "neutral", "contradicted" each, resulting in $91.0\%$, $89.5\%$, $88.0\%$ accuracy. For stochasticity, models may generate $K$ candidate responses per scene, and evaluation uses Best@K: sampling K responses from the RP model and calculate the highest NLI score among recorded outputs. This setup captures both baseline accuracy and the potential best alignment of the role-playing LLM under probabilistic generation. Beyond scripted scenes, we also use `gpt-4.1` as a judge to evaluate methods' performance in open-ended RP scenarios, placed in Appendix H.

**Baselines and Implementations.** We include the following baselines for comparison to show the value of the finite-state machine mechanism inside character state modeling in RP. All methods use the same pair of RP model and the discriminator in all comparisons, and all codification applies `claude-4.0-sonet` with other coding (including open-source) models compared in Appendix E.

- **Vanilla** The LLM receives only the current state and a random action, relying solely on its internal knowledge without explicit transition rules. This forms the lower bound of performance.
- **Textual Profile.** The RP LLM is prompted together with the textual profile of the character.
- **Codified Profile. (state-to-action transition)** (Peng & Shang, 2025) applies codification to represent scene-to-action behavior as executable codes, which take scene texts as input and output grounding behavior statements. It models the logic in a different stage (response inference) in comparison to CFSM/CPFSM (state modeling).
- **PromptTrans. (state-to-state transition)** Instead of executing explicit rules, the LLM is asked to generate the next state as paragraphs from the current state, action, and a textual description of the transition logic. This simulates state progression via summarization, providing a contrast to rule-based execution. The state and scene are further **fed into functions from codified profiles** to get grounded in the state-to-action stage.
- **Character Updating. (state-to-state transition)** (Wang et al., 2025a) maintains an explicit textual *character sheet* that is iteratively rewritten along the trajectory. Given the current state, the latest scene, and the agent's action, the LLM first produces an updated character summary and then generates the response conditioned on this refreshed state. The updated state and scene are further fed into codified-profile functions to obtain grounded behavior in the state-to-action stage.
- **Plot Summary. (state-to-state transition)** (Wang et al., 2025c) maintains a running synopsis of plot-level events shared across turns. At each step, the LLM updates this global plot summary from the previous state, current scene, and action, and uses the revised summary as the new state.

| Artifact | | Haruhi | K-On! | JOJO | FMA | AGOT | ATLA | Average |
|---|---|---|---|---|---|---|---|---|
| | #Character | 5 | 5 | 7 | 5 | 11 | 4 | 5.3 |
| Main | Vanilla (No Profile) | 80.96 | 77.81 | 75.21 | 80.34 | 85.58 | 82.46 | 80.39 |
| | Textual Profile | 81.17 | 80.04 | 75.94 | 80.47 | 85.85 | 83.14 | 81.10 |
| | Codified Profile | 82.77 | 80.49 | 76.19 | 82.58 | 84.47 | 81.66 | 81.36 |
| | PromptTrans | 82.86 | 79.68 | 75.07 | 82.66 | 84.96 | 82.17 | 81.23 |
| | Character Updating | 83.63 | 79.23 | 75.08 | 82.48 | 85.87 | 82.51 | 81.47 |
| | Plot Summary | 83.19 | 79.51 | 76.12 | 82.91 | 85.48 | 83.00 | 81.70 |
| | Codified FSM | 83.88 | 80.45 | 78.31 | 83.89 | 86.11 | 83.28 | **82.65** |
| | #Character | | 4 | 9 | 7 | 19 | 7 | 9.2 |
| Minor | Vanilla (No Profile) | | 80.17 | 78.64 | 83.15 | 87.44 | 82.46 | 82.37 |
| | Textual Profile | | 81.31 | 78.14 | 83.33 | 86.96 | 86.34 | 83.22 |
| | Codified Profile | N/A | 81.54 | 80.58 | 85.91 | 88.91 | 82.16 | 83.82 |
| | PromptTrans | | 82.17 | 81.72 | 83.16 | 86.88 | 82.61 | 83.31 |
| | Character Updating | | 82.13 | 80.39 | 85.49 | 89.13 | 82.57 | 83.59 |
| | Plot Summary | | 82.34 | 80.61 | 85.10 | 88.76 | 82.79 | 83.92 |
| | Codified FSM | | 83.26 | 80.92 | 85.74 | 89.16 | 83.94 | **84.60** |

Table 3: RP performance (NLI Score, default metric without special explanation) comparison with `gpt-4.1` as RP model and `gpt-4.1-mini` as discriminator. N/A: No minor character in Haruhi.

| Artifact | | Method | Haruhi | K-On! | JOJO | FMA | AGOT | ATLA | Average |
|---|---|---|---|---|---|---|---|---|---|
| Merged | `llama-3.2-1b-it` | Textual Profile | 51.60 | 61.64 | 57.87 | 55.42 | 56.35 | 60.10 | 57.16 |
| | | Codified Profile | 55.83 | 63.12 | 57.57 | 60.52 | 59.61 | 64.03 | 60.11 |
| | | Codified FSM | 65.30 | 65.23 | 60.90 | 60.48 | 62.34 | 62.38 | 62.77 |
| | | Codified PFSM | 65.02 | 67.12 | 60.45 | 60.13 | 63.14 | 64.09 | **63.32** |
| | *(w/ Distilled `deberta`)* | Codified Profile | 62.25 | 64.11 | 58.41 | 60.70 | 61.56 | 66.18 | 62.20 |
| | | Codified FSM | 63.50 | 66.12 | 63.90 | 60.95 | 62.49 | 63.25 | 63.37 |
| | | Codified PFSM | 66.74 | 67.41 | 62.93 | 60.51 | 62.93 | 64.32 | **64.14** |

Table 4: 1B LLM and 0.1B distilled discriminator performance (Averaged over all characters.)

The resulting state and scene are then passed to codified-profile functions, which realize the final state-to-action grounding.

- **CFSM/CPFSM. (state-to-state transition)** takes the same input as PromptTrans but updates the state by executing FSM transition codes. FSM states are similarly fed into functions with scenes from codified profiles to get grounded in the state-to-action stage.

## 5.2 MAIN RESULTS

Across both large-scale (`gpt-4.1`) and small-scale (`llama-3.2-1b-instruct`) settings, CFSM consistently achieves the highest performance. In Table 3, CFSM outperforms baselines across all six artifacts, reaching an average of $82.65$ for main characters and $84.60$ for minor characters. Importantly, the widely applied summarization-based state modeling (PromptTrans) does not yield improvements over codified profiles: case analysis in Appendix J shows that it tends to simply repeat surface information already present in the scene, without adding new behavioral cues.

Table 4 further shows that even at the 1B scale, CFSM provides substantial gains, outperforming both textual and codified profiles by large margins. When combined with a small distilled condition checker (by distilling a $0.1B$ `deberta-v3-base` NLI discriminator with $1\%$ discrimination output from `gpt-4.1-mini`, details in Appendix F), CPFSM can further improve upon CFSM with or without distillation. These results highlight CFSM/CPFSM's robustness across model scales and demonstrate that codified FSMs deliver precise state modeling. CPFSM is also found easier to codify than CFSM using weaker coders (e.g., open-source LLMs), shown in Appendix E.

## 5.3 ABLATION AND VARIANT

**Ablation Study.** As shown in Table 5, excluding state registration (always transition from "Unactivated") has the most noticeable impact, showing the importance of state continuity. Removing fallback states like "Other" or "Unactivated" from FSM codification also reduces performance, highlighting their roles in handling uncertain or inactive states. Among FSM types, personality is most critical for guiding action consistency, while identity and ability provide structural support. Together, these elements contribute to CFSM's robustness and its ability to generalize across character types.

| Artifact | | Haruhi | K-On! | JOJO | FMA | AGOT | ATLA | Average |
|---|---|---|---|---|---|---|---|---|
| | Codified FSM | 83.88 | 82.01 | 79.76 | 84.92 | 87.80 | 83.65 | **83.67** |
| Merged | w/o State Registration | 82.31 | 81.44 | 78.33 | 83.65 | 86.96 | 82.79 | 82.58 |
| | w/o "Other" State | 83.24 | 81.47 | 78.95 | 83.78 | 86.88 | 83.19 | 82.92 |
| | w/o "Unactivated" State | 83.62 | 81.53 | 79.58 | 83.93 | 87.01 | 83.35 | 83.17 |
| | w/o Identity FSMs | 81.46 | 82.04 | 78.96 | 84.65 | 87.04 | 82.99 | 82.86 |
| | w/o Personality FSMs | 82.29 | 80.80 | 78.26 | 84.29 | 87.01 | 83.05 | 82.62 |
| | w/o Ability FSMs | 82.02 | 81.99 | 78.47 | 83.77 | 87.42 | 83.08 | 82.79 |

Table 5: Ablation study of the components and FSM types in CFSM.

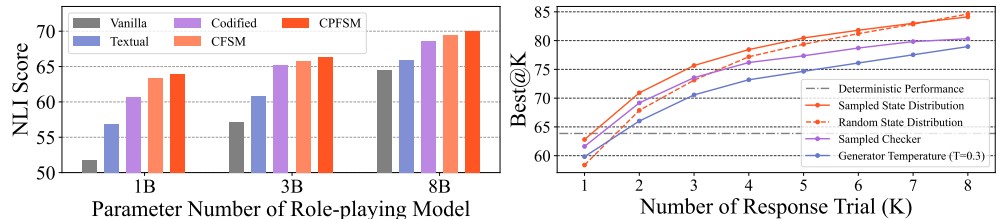

Figure 4: **Left:** CFSM & CPFSM Performance on RP models with different scales. **Right:** Best@K performance of different randomness modeling strategies.

**RP Model Variants.** Using `llama-3-instruct` at 1B, 3B, and 8B parameters, performance rises with scale across all variants, but structurally guided methods dominate at every size. Codified profiles outperform textual prompts and vanilla prompting, and CFSM adds another clear step up, indicating that executing explicit transition logic helps more than simply describing it. CPFSM is best overall and remains ahead even at the smallest scale, showing that lightweight probabilistic checking with logits yields benefits that model size alone does not provide. The gap between structured (CFSM/CPFSM) and unstructured (vanilla/textual) approaches persists as models grow, suggesting that explicit control over state transitions complements scale rather than being replaced by it.

## 5.4 Stochastic Response Exploration

Across all settings shown in the **right** figure in Figure 4, our *Sampled State Distribution* (sampling states from CPFSM distributions) dominates both in sample efficiency and final accuracy: already near the deterministic line at $K=1$, exceeds other baselines by several points, and maintains the highest ceiling as $K$ grows to 7. Simply sampling *Random State Distributions* (complete random state sampling) eventually approaches the curve, but requires many more trials, as a broader exploration with poor guidance. *Sampled Checker* underperforms because sampling yes/no/unknown decisions offers a smaller exploration space. Increasing the *Generator Temperature* (to 0.3) reduces both accuracy and sample efficiency, perturbing surface generation rather than the latent state trajectory. Overall, CPFSM's state-distribution sampling closely tracks deterministic behavior while retaining targeted exploration, yielding the best Best@K performance under comparable compute.

| Method | NLI Score | #Forward/Turn |
|---|---|---|
| Codified Profile | 82.45 | N/A |
| PromptTrans | 81.97 | 130.78 |
| Codified FSM | **83.67** | **30.17** |

(a) Efficiency comparison on execution stage.

| Method | #Question | NLI Score |
|---|---|---|
| Codified Profile | N/A | 62.20 |
| CPFSM ($O(n^2)$) | 50.00 | **64.14** |
| CPFSM ($O(n+k)$) | **17.71** | 63.43 |

(b) Efficiency comparison on codification stage.

Table 6: Efficiency analysis of CFSM and CPFSM.

## 5.5 Efficiency Analysis

**Execution stage.** CFSM maintains the best accuracy while using far fewer forward passes per turn than pure prompting simulation (PromptTrans), because it executes explicit transitions instead of re-reasoning them. Codified Profile incurs no transition-time cost since it does not model internal

| Artifact | | Haruhi | K-On! | JOJO | FMA | AGOT | ATLA | Average |
|---|---|---|---|---|---|---|---|---|
| Merged | Codified FSM | 83.88 | 82.01 | 79.76 | 84.92 | 87.80 | 83.65 | **83.67** |
| | Transition per 2 sentences | 83.56 | 81.66 | 78.78 | 84.21 | 87.32 | 84.48 | 83.33 |
| | Transition per 3 sentences | 82.28 | 80.49 | 78.92 | 84.40 | 87.35 | 84.30 | 82.95 |

Table 7: Investigation of different step lengths for CFSM.

states or perform stepwise updates, but this is also its weakness: without an internal FSM, it cannot enforce consistent multi-step dynamics and trails CFSM in fidelity.

**Codification stage.** We propose an $\mathcal{O}(n+k)$ CPFSM construction to refine efficiency: define $n$ default transition questions for each next state and add only $k$ special-case rules from the profile, avoiding the exhaustive $\mathcal{O}(n^2)$ grid. We implement this $\mathcal{O}(n+k)$ scheme for all characters in the Fandom Benchmark and compare it against the $\mathcal{O}(n^2)$ variant. The approach also reduces the number of discriminative forwards during execution of CPFSM to $O(n + k)$. Furthermore, it preserves accuracy while substantially reducing authoring effort, and the same idea transfers to CFSM by codifying only special transitions and falling back to an $O(n)$ set of default checks when none apply.

### 5.6 FURTHER EXPLORATION

**Step Length Analysis.** To explore the efficiency-accuracy trade-off in CFSM, we evaluate coarser action granularities by assigning one FSM transition per $n$ sentences instead of the default one-per-sentence. As shown in Table 7, although performance decreases slightly with coarser steps, CFSM remains robust: even with 3 sentences per transition, it still outperforms all non-CFSM baselines. This suggests that CFSM can be flexibly adapted to longer update intervals to reduce reasoning overhead, offering a practical speed–accuracy trade-off without sacrificing overall superiority.

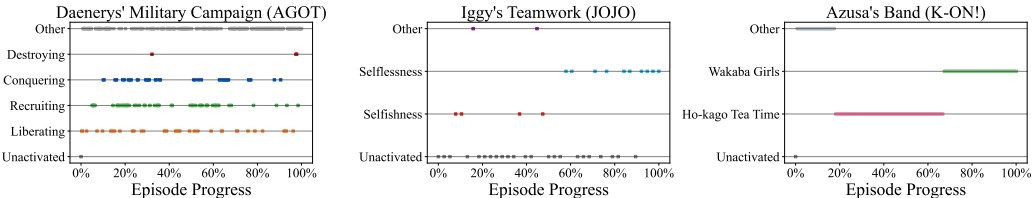

Figure 5: The state dynamics of key states of characters modeled by CFSM.

**Case Study.** CFSM effectively models character dynamics across varied narrative structures. In Game of Thrones, it captures Daenerys' campaign arc through transitions among recruiting, conquering, and liberating, enabling accurate next-action inference based on campaign phase. In JoJo's Bizarre Adventure, CFSM traces Iggy's evolution from selfishness to selflessness, allowing the system to prioritize cooperative actions as his arc progresses. In K-ON!, it identifies Azusa's shift in band identity, constraining likely actions to group-centric behaviors once her membership state updates. These examples highlight CFSM's ability to surface latent state trajectories and guide coherent next steps, which is something prompt-only approaches fail to robustly support.

### 6 CONCLUSION AND FUTURE WORK

We presented Codified Finite-State Machines (CFSM) and their probabilistic extension (CPFSM) as interpretable frameworks for modeling character state transitions, improving coherence and efficiency over prompting-based baselines. Future work will focus on three key directions: (1) automatically building CFSMs directly from narrative plots rather than pre-written profiles, (2) extending CFSMs with numeric mechanisms such as health points to capture continuous dynamics, and (3) enabling dynamic updates to the state set, allowing characters to acquire new skills or traits over time. These directions aim to make CFSMs more adaptive and closer to the evolving logic of real narratives.

ETHICS STATEMENT

This work focuses on improving the coherence and interpretability of role-playing systems. All datasets are curated from publicly available Fandom sources, and no private or sensitive information is used. The proposed methods are intended for research in controllable character modeling and should not be misused for generating harmful or deceptive content.

REPRODUCIBILITY STATEMENT

We provide detailed implementation steps, codification prompts, and evaluation setups in the appendices. All experiments are conducted with specified models, datasets, and hyperparameters to ensure replicability. The codebase and processed benchmark data will be released to support full reproducibility.

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

## A  USE OF LLMS

In this paper, large language models are employed in a strictly limited and transparent manner. Their role is confined to two specific purposes: (1) light polishing of the writing style to improve clarity and readability, and (2) execution of controlled experiments in the sections where the methodology explicitly specifies the usage of an LLM. Beyond these stated scenarios, no hidden or implicit reliance on LLMs is made, ensuring that the conceptual contributions, analyses, and results presented in the work remain independent of automated language model generation.

## B  LIMITATION AND FUTURE WORK

While CFSM and CPFSM provide interpretable and efficient mechanisms for modeling character states in role-playing, several limitations remain that point to promising directions for future research. These limitations stem from simplifying assumptions designed for tractability and computational efficiency, which may restrict expressiveness in complex narrative environments.

First, the framework assumes a fixed set of states throughout an episode, which limits the ability to represent emergent or evolving traits such as a character gaining a new skill. Future work could enable dynamic state construction by adding, merging, or redefining states as the plot progresses, while maintaining coherent transition logic.

Second, `binary_question` evaluations are currently performed in a single forward pass, which may be insufficient for harder semantic conditions. A future improvement is to introduce a gating mechanism, for example a `thinking` variable, that triggers deeper reasoning such as multi-step inference or fallback to a larger model when the question requires context-dependent interpretation.

Third, CFSM assumes Markovian transitions that depend only on the current state and immediate input. This simplifies execution but cannot capture behaviors that depend on past outcomes. Future work could support non-Markovian transitions by incorporating episodic memory, such as modeling reluctance to reuse a failed ability or enforcing cooldown periods, which can be codified as higher-order conditions with temporal depth.

Taken together, these directions suggest a path toward more adaptive and expressive state modeling. By supporting dynamic state growth, adaptive reasoning, and non-Markovian transitions, future CFSM variants can capture richer and longer-term character behavior in open-ended role-playing environments.

## C  PROMPTS, TEMPLATES, AND CODES

From Figure 6 to 10, we list the prompts and (FSM) codes used in our experiments for codification, checking, and evaluation. For prompts in baseline methods, we reuse exactly the same prompts in the previous codified profile work (Peng & Shang, 2025) for reproduction.

## D  STATISTICS OF FSM CODIFICATION

As shown in Table 8, the statistics outline structural and density features across main and minor artifacts. For main groups, CFSM counts range from lower values around 11 to higher values above 20, paired with paragraph counts generally in the low to mid-teens. Vertex averages tend to cluster around 4.5–5.3, while edge averages sit between about 15 and 19, producing density values that mostly hover in the mid-0.7 range, with one case approaching 0.85. Minor groups show similar distributions but with more variation: CFSM spans from around 11 to over 20, paragraph counts from about 7 to 15, vertex values from 4.7 to just above 5, and edges generally between 16 and 18.5. Density in the minors shows a broader spread, with values from roughly 0.67 up to over 0.81. Together, these figures capture the balance between structure size (CFSM, paragraphs), connectivity (edges, vertices), and compactness (density) in each artifact category.

## Prompt #1: Extract Key States

```
# Character

{character} (always referring as this name)

# Profile

{profile}

# Task

Given the character profile above of {character}, propose potential finite-state machines (FSMs) for character states that can be der
ived to generally apply for many scenes.
For each finite-state machine, write a list of potential **mutually exclusive** and **distinct** states (not actions) in the finite-s
tate machine that are explicitly described by the profile.
Each finite-state machine must have an unactivated state (first of the list) and an other state (last of the list, **explicitly menti
on other than what**) to cover all kinds of situations.
Finally, summarize the states in FSMs as a dictionary (FSM name: list of states) of lists of sentences mentioning "{character}":

```python
fsm_states = {{
# FSM1
"...": [..., ...., ],
# FSM2
"...": [..., ...., ],
...
}}
```
```

## Prompt #2: Codify into CFSM

```
# Character

{character} (always referring as this name)

# Profile

{profile}

# FSM States

{states}

Based on the given profile, build the transition function for the FSM states.
- Input: `current_state`, a string contains a state inside the given FSM states;
`scene`, a string describing the current scene;
`action`, a string describing an action taken by a character in the current scene;
`active`, a bool for whether it's {character} (`True`) taking the `action` or not (False).
- Output: `next_state`, a string containing a state inside the given FSM states, should always be indexed by `next_state_idx`;
`triggered_extra_states`, a list of string containing extra 1-sentence states that should be triggered by the transition according
to the profile.
- states in `triggered_extra_states` should also be syntactically complete, independent on reference.

You can use a pre-defined helper function `binary_question(text, question)` to check the inputs to build the logic. `binary_questio
n` returns `True`/`False`/`None` (Unknown) for if logic.
- `binary_question` can be applied for both `scene` and `action` with clear `question` with resolved ambiguity and coreference.
- `question` should be a syntactically complete question, independent on previous questions.

Complete the TASK above step by step:

1. Explain the state transition logic in the given FSM states indicated by the given profile.

2. Draw two state transition graphs (active and passive) that reflects nested `if` and `if-else` relations.

3. Search for ALL potential triggered extra states from the given profile and analyze their situations for triggering. (`if` and `i
f-else` relations)

4. Compile the character profile into the state transition function `fsm_transition(current_state, scene, action, active)`.

# Expected format:

```python
def fsm_transition(current_state, scene, action, active):

    all_states = {states}
    triggered_extra_states = []

    if active:
        # The action is taken by {character}

        if current_state == all_states[0]:
            # TODO (get next_state_idx, append triggered states to triggered_extra_states)

        if current_state == all_states[1]:
            # TODO (get next_state_idx, append triggered states to triggered_extra_states)

        # TODO

    else:
        # The action is taken by another character

        if current_state == all_states[0]:
            # TODO (get next_state_idx, append triggered states to triggered_extra_states)

        if current_state == all_states[1]:
            # TODO (get next_state_idx, append triggered states to triggered_extra_states)

        # TODO

    next_state = all_states[next_state_idx]

    return next_state, triggered_extra_states
```
```

Figure 6: The prompts and codes used in our experiments (1/5).

**Prompt #3: Codify into CPFSM**

```
# {full_name}'s Character Logic

{profile}

# Source State (State {idx})

{state}

# Target States

{states}

# Task

Based on {full_name}'s profile, generate lists of questions to check whether the source state should transition to each target state w
hen a new action is taken.
The "active" list ({len(states)} questions) is for the situation that the action is taken by {full_name};
The "passive" list ({len(states)} questions) is for the situation that the action is taken by another character.

Step 1. Reasoning about the transition logic.

Reasoning: ...

Step 2. Output the question lists

Expected format: (No comments)
```json
{{
    "active": <a list of {len(states)} strings>,
    "passive": <a list of {len(states)} strings>,
}}
```
```

**Prompt #4: Relevance Check**

```
# Scene:
{scene_segment}

# Action:
{action}

# Question:

In the given scene, does the action contain any character's active action AND with {character_abbr} ({character}) in the current (not h
istory) scene?

Directly answer only yes/no/unknown.
```

**Prompt #5: Active/Passive Check**

```
# Scene:
{scene_segment}

# Action:
{action}

# Question:

In the given scene, is the action taken by {character_abbr} ({character})?

Directly answer only yes/no/unknown.
```

**Prompt #6: Generate Respond**

```
# Background Knowledge
{grounding}
{triggered}

# Scene
{scene}

# Question
{question} Answer a concise narration in one sentence.
```

**Prompt #7: NLI Scoring**

```
# Scene
{scene}

Your Response: {prediction}
Ground Truth: {reference}

Score the action of {character_abbr} in the response based on the ground truth.
A: The ground truth entails the action of {character_abbr} in the response. (Following the same character logic.)
B: The ground truth is neutral to the action of {character_abbr} in the response. (Reflecting a different facet.)
C: The ground truth contradicts the action of {character_abbr} in the response. (Following a contradicted character logic.)

Output in json:
```json
{
"reasoning": "...",
"score": "A/B/C"
}
```
```

Figure 7: The prompts and codes used in our experiments (2/5).

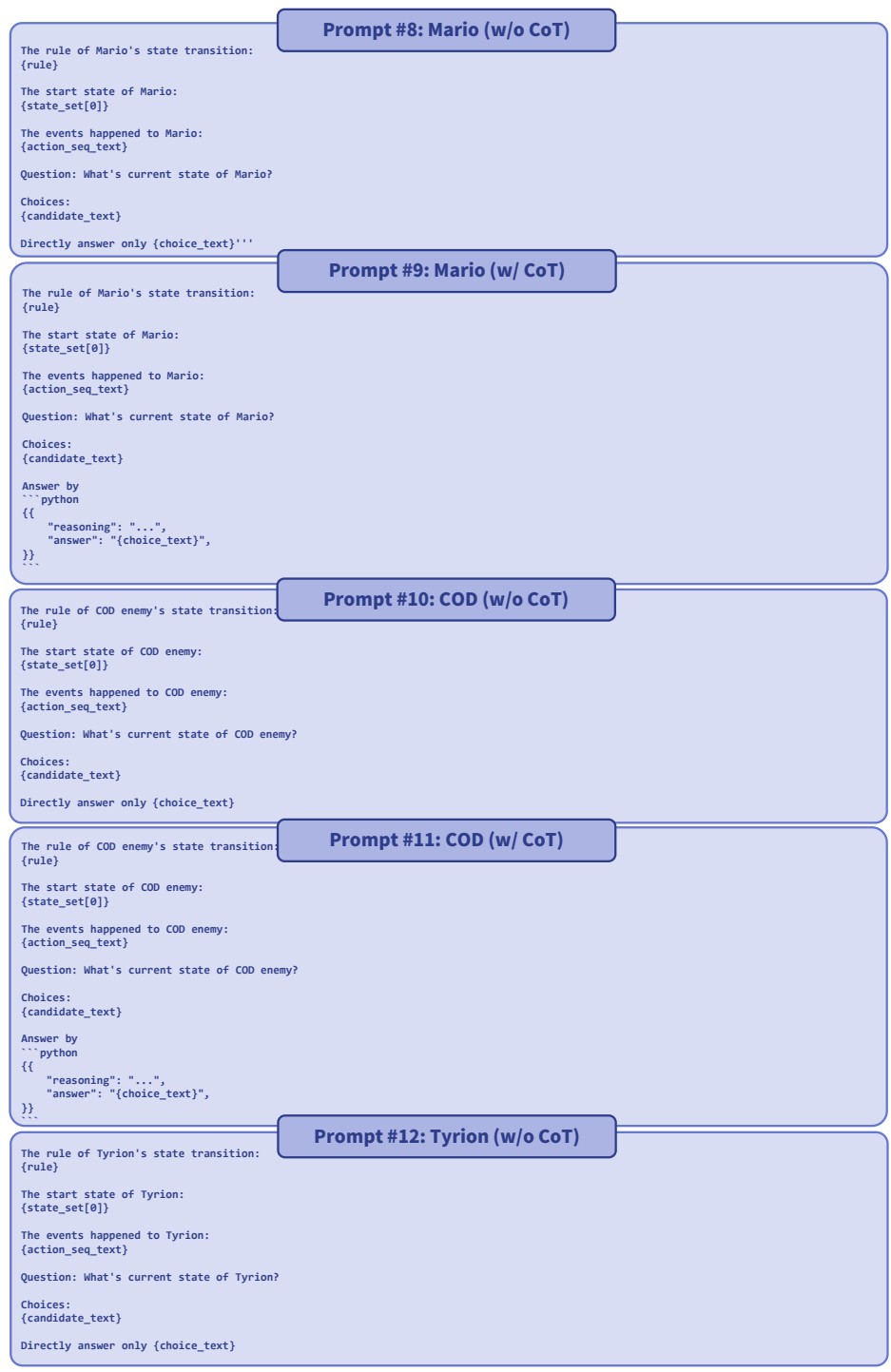

**Prompt #8: Mario (w/o CoT)**

```
The rule of Mario's state transition:
{rule}

The start state of Mario:
{state_set[0]}

The events happened to Mario:
{action_seq_text}

Question: What's current state of Mario?

Choices:
{candidate_text}

Directly answer only {choice_text}'''
```

**Prompt #9: Mario (w/ CoT)**

```
The rule of Mario's state transition:
{rule}

The start state of Mario:
{state_set[0]}

The events happened to Mario:
{action_seq_text}

Question: What's current state of Mario?

Choices:
{candidate_text}

Answer by
```python
{{
    "reasoning": "...",
    "answer": "{choice_text}",
}}
```
```

**Prompt #10: COD (w/o CoT)**

```
The rule of COD enemy's state transition:
{rule}

The start state of COD enemy:
{state_set[0]}

The events happened to COD enemy:
{action_seq_text}

Question: What's current state of COD enemy?

Choices:
{candidate_text}

Directly answer only {choice_text}
```

**Prompt #11: COD (w/ CoT)**

```
The rule of COD enemy's state transition:
{rule}

The start state of COD enemy:
{state_set[0]}

The events happened to COD enemy:
{action_seq_text}

Question: What's current state of COD enemy?

Choices:
{candidate_text}

Answer by
```python
{{
    "reasoning": "...",
    "answer": "{choice_text}",
}}
```
```

**Prompt #12: Tyrion (w/o CoT)**

```
The rule of Tyrion's state transition:
{rule}

The start state of Tyrion:
{state_set[0]}

The events happened to Tyrion:
{action_seq_text}

Question: What's current state of Tyrion?

Choices:
{candidate_text}

Directly answer only {choice_text}
```

Figure 8: The prompts and codes used in our experiments (3/5).

**Prompt #13: Tyrion (w/ CoT)**

```
The rule of Tyrion's state transition:
{rule}

The start state of Tyrion:
{state_set[0]}

The events happened to Tyrion:
{action_seq_text}

Question: What's current state of Tyrion?

Choices:
{candidate_text}

Answer by the following format
```json
{{
    "reasoning": "...",
    "answer": "{choice_text}",
}}
```
```

**Code #1: Mario**

```python
def get_next_state(state, action):
    if state == "small mario":
        if binary_question(action, "Does Mario get a super mushroom?"):
            return "super mario"
        if binary_question(action, "Is Mario hit by a Goomba?"):
            return "miss"
    elif state == "super mario":
        if binary_question(action, "Does Mario get a fire flower?"):
            return "fire mario"
        if binary_question(action, "Is Mario hit by a Goomba?"):
            return "small mario"
    elif state == "fire mario":
        if binary_question(action, "Is Mario hit by a Goomba?"):
            return "super mario"
    return state
```

**Code #2: COD**

```python
def get_next_state(state, action):
    if state == "idle":
        if binary_question(action, "Does the enemy hear a suspicious sound or see a hint of the player?"):
            return "alert"
        if binary_question(action, "Is the enemy killed unexpectedly (e.g., stealth takedown)?"):
            return "death"
    elif state == "alert":
        if binary_question(action, "Does the enemy positively confirm the player's position?"):
            return "engaged"
        if binary_question(action, "Is the suspicion unresolved and the alert timer expires?"):
            return "idle"
        if binary_question(action, "Is the enemy killed during alert state?"):
            return "death"
    elif state == "engaged":
        if binary_question(action, "Is the enemy under heavy fire or low on health?"):
            return "retreat"
        if binary_question(action, "Does the enemy lose sight of the player for long enough?"):
            return "alert"
        if binary_question(action, "Is the enemy killed in combat?"):
            return "death"
    elif state == "retreat":
        if binary_question(action, "Has the enemy recovered and re-established contact with the player?"):
            return "engaged"
        if binary_question(action, "Is the enemy safe and has lost contact with the player?"):
            return "alert"
        if binary_question(action, "Is the enemy killed while retreating?"):
            return "death"
    elif state == "death":
        return "death"  # terminal
    # No transition triggered -> stay in current state
    return state
```

**Code #3: Tyrion (1/2)**

```python
def get_next_state(state, action):

    if state == "the north":
        if binary_question(action, "Does Tyrion travel south (exactly south, not southeast or southwest)?"):
            return "the riverlands"
        if binary_question(action, "Does Tyrion travel southeast (exactly southeast, not just south or east)?"):
            return "the vale"
        if binary_question(action, "Does Tyrion travel southwest (exactly southwest, not just south or west)?"):
            return "the westerlands"

    elif state == "the vale":
        if binary_question(action, "Does Tyrion travel west (exactly west, not northwest or southwest)?"):
            return "the riverlands"
        if binary_question(action, "Does Tyrion travel south (exactly south, not southeast or southwest)?"):
            return "the crownlands"
        if binary_question(action, "Does Tyrion travel southeast (exactly southeast, not just south or east)?"):
            return "the stormlands"

    elif state == "the riverlands":
        if binary_question(action, "Does Tyrion travel north (exactly north, not northeast or northwest)?"):
            return "the north"
        if binary_question(action, "Does Tyrion travel east (exactly east, not northeast or southeast)?"):
            return "the vale"
        if binary_question(action, "Does Tyrion travel west (exactly west, not northwest or southwest)?"):
            return "the westerlands"
        if binary_question(action, "Does Tyrion travel southeast (exactly southeast, not just south or east)?"):
            return "the crownlands"
        if binary_question(action, "Does Tyrion travel southwest (exactly southwest, not just south or west)?"):
            return "the reach"
```

Figure 9: The prompts and codes used in our experiments (4/5).

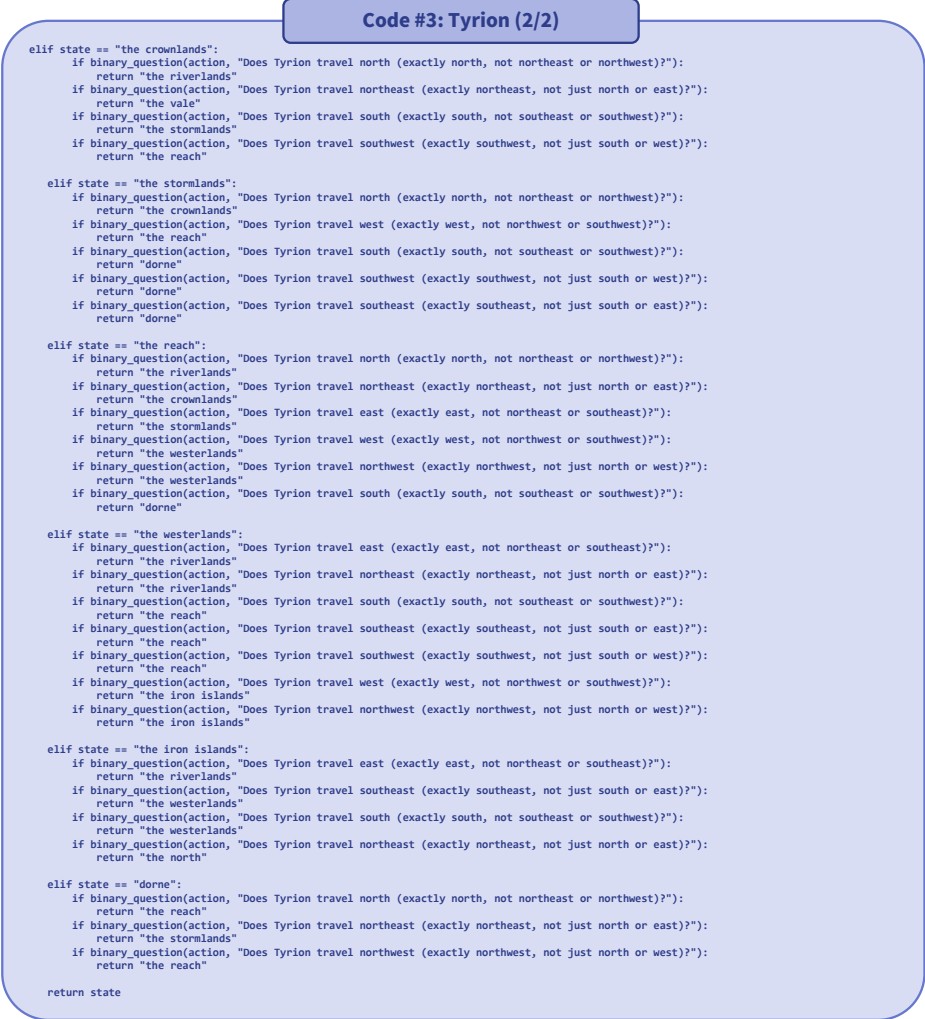

```
                        Code #3: Tyrion (2/2)

elif state == "the crownlands":
    if binary_question(action, "Does Tyrion travel north (exactly north, not northeast or northwest)?"):
        return "the riverlands"
    if binary_question(action, "Does Tyrion travel northeast (exactly northeast, not just north or east)?"):
        return "the vale"
    if binary_question(action, "Does Tyrion travel south (exactly south, not southeast or southwest)?"):
        return "the stormlands"
    if binary_question(action, "Does Tyrion travel southwest (exactly southwest, not just south or west)?"):
        return "the reach"

elif state == "the stormlands":
    if binary_question(action, "Does Tyrion travel north (exactly north, not northeast or northwest)?"):
        return "the crownlands"
    if binary_question(action, "Does Tyrion travel west (exactly west, not northwest or southwest)?"):
        return "the reach"
    if binary_question(action, "Does Tyrion travel south (exactly south, not southeast or southwest)?"):
        return "dorne"
    if binary_question(action, "Does Tyrion travel southwest (exactly southwest, not just south or west)?"):
        return "dorne"
    if binary_question(action, "Does Tyrion travel southeast (exactly southeast, not just south or east)?"):
        return "dorne"

elif state == "the reach":
    if binary_question(action, "Does Tyrion travel north (exactly north, not northeast or northwest)?"):
        return "the riverlands"
    if binary_question(action, "Does Tyrion travel northeast (exactly northeast, not just north or east)?"):
        return "the crownlands"
    if binary_question(action, "Does Tyrion travel east (exactly east, not northeast or southeast)?"):
        return "the stormlands"
    if binary_question(action, "Does Tyrion travel west (exactly west, not northwest or southwest)?"):
        return "the westerlands"
    if binary_question(action, "Does Tyrion travel northwest (exactly northwest, not just north or west)?"):
        return "the westerlands"
    if binary_question(action, "Does Tyrion travel south (exactly south, not southeast or southwest)?"):
        return "dorne"

elif state == "the westerlands":
    if binary_question(action, "Does Tyrion travel east (exactly east, not northeast or southeast)?"):
        return "the riverlands"
    if binary_question(action, "Does Tyrion travel northeast (exactly northeast, not just north or east)?"):
        return "the riverlands"
    if binary_question(action, "Does Tyrion travel south (exactly south, not southeast or southwest)?"):
        return "the reach"
    if binary_question(action, "Does Tyrion travel southeast (exactly southeast, not just south or east)?"):
        return "the reach"
    if binary_question(action, "Does Tyrion travel southwest (exactly southwest, not just south or west)?"):
        return "the reach"
    if binary_question(action, "Does Tyrion travel west (exactly west, not northwest or southwest)?"):
        return "the iron islands"
    if binary_question(action, "Does Tyrion travel northwest (exactly northwest, not just north or west)?"):
        return "the iron islands"

elif state == "the iron islands":
    if binary_question(action, "Does Tyrion travel east (exactly east, not northeast or southeast)?"):
        return "the riverlands"
    if binary_question(action, "Does Tyrion travel southeast (exactly southeast, not just south or east)?"):
        return "the westerlands"
    if binary_question(action, "Does Tyrion travel south (exactly south, not southeast or southwest)?"):
        return "the westerlands"
    if binary_question(action, "Does Tyrion travel northeast (exactly northeast, not just north or east)?"):
        return "the north"

elif state == "dorne":
    if binary_question(action, "Does Tyrion travel north (exactly north, not northeast or northwest)?"):
        return "the reach"
    if binary_question(action, "Does Tyrion travel northeast (exactly northeast, not just north or east)?"):
        return "the stormlands"
    if binary_question(action, "Does Tyrion travel northwest (exactly northwest, not just north or west)?"):
        return "the reach"

return state
```

Figure 10: The prompts and codes used in our experiments (5/5).

| | Artifact | Haruhi | K-On! | JOJO | FMA | AGOT | ATLA |
|---|---|---|---|---|---|---|---|
| **Main** | $n_{\text{cfsm}}$ | 11.60 | 14.60 | 17.57 | 20.00 | 20.73 | 23.50 |
| | $n_{\text{paragraph}}$ | 11.80 | 9.20 | 15.71 | 16.00 | 14.64 | 14.50 |
| | $n_v$ | 4.67 | 4.70 | 4.27 | 4.88 | 5.05 | 5.29 |
| | $n_e$ | 17.02 | 16.84 | 15.47 | 18.38 | 17.71 | 19.35 |
| | Density ($\frac{n_e}{n_v^2}$) | 0.78 | 0.76 | 0.85 | 0.78 | 0.70 | 0.69 |
| **Minor** | $n_{\text{cfsm}}$ | | 11.25 | 11.33 | 20.86 | 17.05 | 13.57 |
| | $n_{\text{paragraph}}$ | | 7.00 | 9.89 | 15.14 | 11.84 | 9.29 |
| | $n_v$ | N/A | 4.73 | 4.67 | 5.01 | 5.19 | 4.96 |
| | $n_e$ | | 18.28 | 16.24 | 18.45 | 18.13 | 18.64 |
| | Density ($\frac{n_e}{n_v^2}$) | | 0.82 | 0.76 | 0.73 | 0.67 | 0.76 |

Table 8: Descriptive statistics of artifacts (CFSM count, paragraph count, average vertices $n_v$, edges $n_e$, and density). N/A: No minor character in Haruhi.

| Artifact | | Model | Haruhi | K-On! | JOJO | FMA | AGOT | ATLA | Average |
|---|---|---|---|---|---|---|---|---|---|
| Main | CFSM | `claude-4.0` | 63.50 | 65.21 | 58.27 | 62.23 | 61.13 | 61.94 | 62.04 |
| | | `gpt-4.1` | 61.07 | 66.40 | 60.26 | 61.62 | 61.58 | 61.05 | 62.00 |
| | | `qwen3-coder` | 60.25 | 65.46 | 56.38 | 60.35 | 60.07 | 60.83 | 60.56 |
| | CPFSM | `claude-4.0` | 66.74 | 67.20 | 57.97 | 61.78 | 60.46 | 62.87 | 62.84 |
| | | `gpt-4.1` | 65.89 | 66.79 | 59.03 | 60.83 | 61.22 | 61.49 | 62.54 |
| | | `qwen3-coder` | 67.62 | 64.92 | 58.64 | 62.37 | 60.43 | 63.27 | 62.88 |
| Minor | CFSM | `claude-4.0` | N/A | 65.02 | 63.82 | 58.33 | 64.98 | 63.12 | 63.06 |
| | | `gpt-4.1` | | 65.31 | 66.48 | 56.61 | 64.02 | 62.55 | 62.99 |
| | | `qwen3-coder` | | 64.38 | 62.21 | 57.05 | 63.75 | 62.29 | 61.94 |
| | CPFSM | `claude-4.0` | N/A | 67.58 | 66.92 | 59.48 | 64.91 | 65.48 | 64.88 |
| | | `gpt-4.1` | | 66.83 | 64.85 | 58.94 | 64.69 | 66.13 | 64.29 |
| | | `qwen3-coder` | | 65.68 | 65.94 | 60.90 | 64.25 | 64.89 | 64.33 |

Table 9: Comparison of codification models for implementing CFSM and CPFSM.

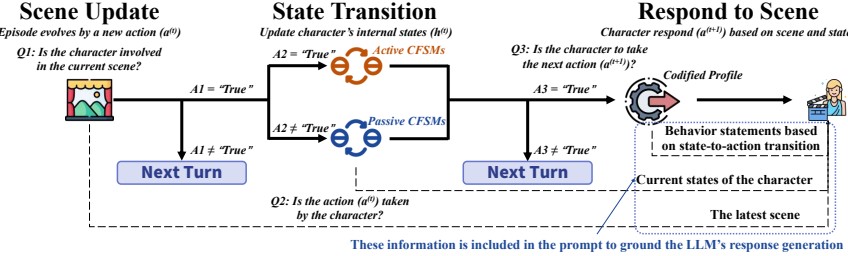

Figure 11: A more specific workflow implementation of CFSM/CPFSM in our experiments.

# E CODIFICATION MODEL COMPARISON

As shown in Table 9, including state-of-the-art closed-source LLMs (`claude-4.0-sonnet`) and an open-source LLM (`qwen3-coder-plus`, a 480B coding model). The closed-source LLMs (`gpt-4.1` and `claude-4.0-sonnet`) outperform in the codification of deterministic CFSMs, where accurate generation of structured code and conditional logic is essential. These models consistently produce cleaner, more reliable transition functions with fewer errors in control flow and state definitions. This reflects their stronger general-purpose coding capabilities and better instruction-following behavior, which are critical for constructing full FSM implementations.

In contrast, the performance gap narrows for CPFSM codification, which primarily involves generating semantically meaningful transition questions rather than writing executable code. On this task, the open-source model `qwen3-coder-plus` shows competitive results, suggesting that high-quality question generation is less dependent on expert-level coding ability and more on semantic understanding. This indicates CPFSM may be more accessible to mid-scale models and suitable for deployment in low-resource or open-source environments.

# F FURTHER IMPLEMENTATION DETAILS

Figure 11 plots a more detailed implementation of the CFSM/CPFSM pipeline, which begins by parsing each plot turn into a scene–action tuple, which is passed through the distilled discriminator to handle three checks for every character: relevance (whether the character is affected), activity (active vs. passive role), and condition evaluation for `binary_question`. Irrelevant characters are skipped, while relevant ones are routed to their active or passive CFSM/CPFSM. In CFSM, the codified `get_next_state` function combines default guards with special rules extracted from the profile, calling `binary_question` when needed to determine the next state. In CPFSM, the same questions provide logits for a transition row, which updates the character's state distribution; the most likely state grounds the role-play while the full distribution is preserved for uncertainty tracking. After states are updated, the state-to-action step is performed by the small discriminator following

| Artifact | | Haruhi | K-On! | JOJO | FMA | AGOT | ATLA | Average |
|---|---|---|---|---|---|---|---|---|
| **Main** | #Character | 5 | 5 | 7 | 5 | 11 | 4 | 5.3 |
| | Codified Profile | 20.57 | 18.50 | 18.82 | 18.82 | 20.81 | 18.95 | 19.41 |
| | PromptTrans | 20.09 | 18.09 | 18.72 | 18.91 | 20.60 | 18.68 | 19.18 |
| | Character Updating | 20.71 | 18.05 | 18.53 | 18.58 | 20.10 | 18.81 | 19.13 |
| | Plot Summary | 21.11 | 18.22 | 17.98 | 18.93 | 20.52 | 18.51 | 19.21 |
| | Codified FSM | 20.61 | 19.09 | 18.78 | 19.32 | 21.28 | 19.35 | **19.74** |
| **Minor** | #Character | | 4 | 9 | 7 | 19 | 7 | 9.2 |
| | Codified Profile | | 21.21 | 18.09 | 21.40 | 21.66 | 19.17 | 20.31 |
| | PromptTrans | | 20.18 | 18.13 | 20.78 | 21.27 | 19.06 | 19.88 |
| | Character Updating | N/A | 19.51 | 18.39 | 21.61 | 20.81 | 18.87 | 19.84 |
| | Plot Summary | | 20.19 | 18.38 | 21.16 | 21.67 | 18.93 | 20.07 |
| | Codified FSM | | 20.93 | 18.74 | 22.88 | 21.49 | 20.02 | **20.81** |

Table 10: RP performance comparison with `gpt-4.1` as the RP model and `gpt-4.1-mini` as the discriminator with **ROUGE-L** as the evaluation metric.

the codified profile, mapping the current state and scene into a concrete action behavior. Finally, the scene, state, and action are appended to the role-playing prompt, guiding the larger RP model in generating the character's utterance or behavior.

**Distillation Fine-turning**   To improve efficiency across all classification components, we employ a unified distilled discriminator for three tasks: `binary_question` used in `get_next_state`, character relevance checking, and active/passive role classification. Rather than querying the role-playing model repeatedly, we distill a compact model from the outputs of `gpt-4.1-mini`. Specifically, we randomly sample 1% of the discrimination cases generated during CFSM experiments. A 3-class `deberta-v3-base` model (0.1B) (He et al., 2021) is trained for 5 epochs on 90% of the data and achieves 79.1% consistency with the teacher model on the held-out 10%. This lightweight checker enables scalable and fast evaluation of condition logic throughout the CFSM pipeline.

## G   MODEL-INDEPENDENT EVALUATION

Table 10 reports complementary results using ROUGE-L (Lin, 2004) to provide a model-independent perspective on RP quality following previous RP measurement (Wang et al., 2025c). Because ROUGE measures lexical alignment rather than relying on an LLM discriminator, it offers an orthogonal view of consistency across trajectories. Under this metric, CFSM continues to exhibit strong performance across both main and minor characters, suggesting that its state-structured transitions support stable behavior even when evaluated without generative-model judgments.

## H   OPEN-ENDED RP EVALUATION

To assess how codified approaches perform outside scripted benchmarks, we construct 200 interactive scenes spanning activities such as casual play, combat encounters, and negotiation tasks. From each initial setup, we simulate roleplay for 10 consecutive turns, with actions sampled at each step according to the method being tested. The resulting trajectories are then compared pairwise between CFSM/CPFSM and codified profile baselines. Evaluation is conducted by `gpt-4.1` acting as a judge, which reviews

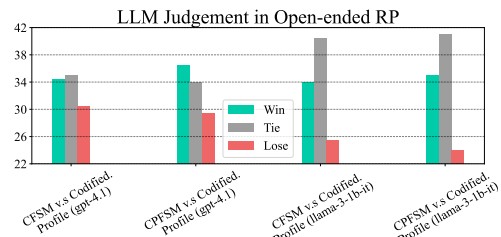

Figure 12: Comparison of open-ended RP by LLM judgment.

the two continuations and determines whether one approach better preserves character logic and consistency, whether they are comparable, or whether one fails relative to the other.

Results in Figure 12 show that CFSM and CPFSM consistently achieve higher win rates over codified profiles, with notably fewer losses. This advantage is especially evident when evaluated with smaller RP models, where codified profiles often drift or hallucinate without explicit state tracking. These

| Method | Textual Profile | Text + CoT | Text + CoT + In-Context States | CFSM |
|---|---|---|---|---|
| Correct Interaction (%) | 88.72 | 92.36 | 95.75 | **96.13** |
| Extra Fowarding per Round | - | 73.84 | 35.13 | **1.00** |

Table 11: Social interaction simulation performance.

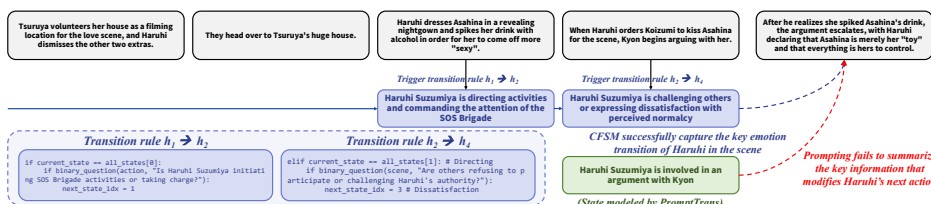

Figure 13: A case study on how CFSM precisely models key state transition.

findings confirm that codified FSMs not only excel in controlled benchmarks but also provide tangible benefits in open-ended, multi-turn role-play, where long-term coherence and traceable logic are critical.

## I    SOCIAL INTERACTION SIMULATION

To evaluate the robustness of CFSM in realistic conversational scenarios, we design a social-interaction evaluation that spans *five roles (Customer Support Agent, Medical Triage Nurse, Teacher/Tutor, Interviewer, and Project Manager)*. For each role, we instantiate *ten distinct test personas* (e.g., *impatient customer*) representing diverse communication styles and goals, resulting in a broad set of social contexts. Each simulated conversation proceeds as a *goal-driven, multi-round interaction* capped at 20 turns, during which the agent must navigate evolving user intent, latent emotional cues, and role-specific behavioral constraints.

Across five roles and ten diverse personas, we evaluate models in goal-driven, multi-round (20 turns at most, based on gpt-4.1) social interactions using two metrics: *Correct Interaction*, the percentage of **turns** that follow the role manual, and *Extra Forwarding per Round*, the average number of additional reasoning or filler tokens required to produce a stable response. As shown in Table 11, textual profiles alone provide decent but imperfect consistency, while adding CoT improves correctness at the cost of substantial extra forwarding due to repeated re-derivation of latent states. Supplying in-context states mitigates this overhead by letting the prompting process simulate an FSM, though at the expense of longer prompts. In contrast, CFSM yields the highest correctness and the lowest forwarding cost because its explicit, compact transition structure allows the model to execute state updates directly rather than infer them anew each turn. Overall, CFSM offers superior behavioral fidelity and computational efficiency for multi-turn social interaction simulation.

## J    EXTENDED CASE STUDY

Figure 13 illustrates how CFSM captures Haruhi Suzumiya's key state transitions during a film-shoot scene. When Haruhi initiates the SOS Brigade activity, the codified rule $h_1 \to h_2$ is triggered by checking *"Is Haruhi initiating SOS Brigade activities or taking charge?"*, moving her into a *directing/commanding* state. Later, when Kyon resists her orders and exposes the spiked drink, the rule $h_2 \to h_4$ fires through *"Are others refusing to participate or challenging Haruhi's authority?"*, shifting her into a *confrontational* state. These explicit checks ensure that decisive cues, like opposition to her authority are registered, producing a transparent and reproducible state update.

In contrast, prompt-only continuation tends to summarize surface events and overlook the turning point that drives Haruhi's change in behavior. By codifying transitions as explicit rules and recording each outcome, CFSM provides clear reasoning traces that ground the RP model's next action, ensuring coherent and faithful role-play.

Figure 14 shows three examples of chain-of-thoughts: Mario's item interactions, an enemy's stealth logic in *Call of Duty*, and Tyrion's regional travels. In each case, chain-of-thought prompting correctly

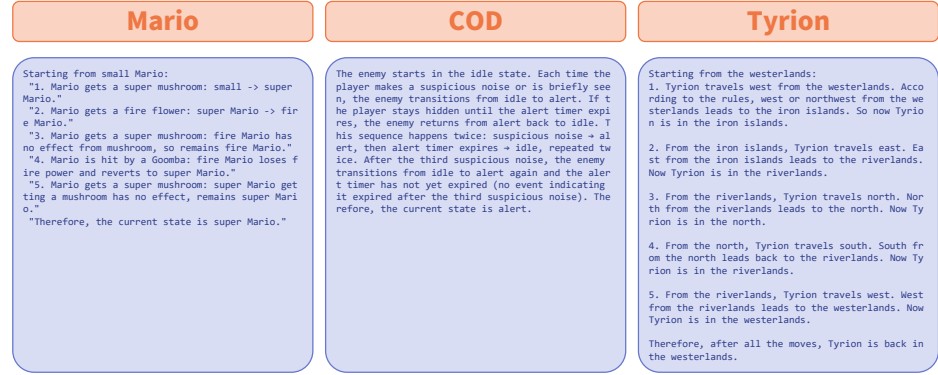

Figure 14: Cases of chain-of-thoughts as simulating FSMs in LLMs solving state transition.

| Artifact | | Haruhi | K-On! | JOJO | FMA | AGOT | ATLA | Average |
|----------|------|--------|-------|------|------|------|------|---------|
| Main | Codified FSM | 83.88 | 80.45 | 78.31 | 83.89 | 86.11 | 83.28 | 82.65 |
| | + Dynamic States | 84.83 | 81.78 | 78.02 | 84.16 | 85.79 | 83.64 | 83.04 |
| Minor | Codified FSM | N/A | 83.26 | 80.92 | 85.74 | 89.16 | 83.94 | 84.60 |
| | + Dynamic States | | 82.97 | 80.27 | 85.36 | 88.79 | 85.02 | 84.48 |

Table 12: RP performance with dynamic state maintenance mechanism.

follows the transition rules and produces the right final state, demonstrating that LLMs are capable of mimicking explicit FSM behavior. However, this correctness comes at a steep cost: CoT requires roughly 25 times more forward passes than CFSM to reach the same outcome. By contrast, CFSM executes the codified rules directly, yielding identical results with far greater efficiency. These examples highlight that while LLMs can simulate FSM reasoning in principle, codified FSMs achieve the same fidelity with orders-of-magnitude less computation and with transparent transition traces.

## K  DYNAMIC STATE MAINTENANCE

The CFSM framework assumes a finite, codified set of character states extracted from profiles. In practice, however, long-form RP naturally gives rise to *emergent* or highly specific states that were not anticipated during codification (e.g., rare emotional combinations or situation-specific role shifts). In the base CFSM, such cases are absorbed into a generic `other` state, which preserves correctness but under-specifies the character's fine-grained dynamics. To explore how CFSM can flexibly accommodate these emergent states, we introduce a *Dynamic State Maintenance* mechanism as an additional layer on top of the codified FSM.

Concretely, we augment each CFSM with a dynamically maintained list of refined `other`-type substates. When a transition leads to the generic `other` state, the LLM inspects the current trajectory and selects a more specific label from the maintained list. During evaluation, whenever a reference action is observed and the CFSM would fall into `other`, the LLM is asked to clarify what that `other` state should be (e.g., "reserved but reluctantly cooperative" vs. "openly confrontational but conflicted") and to add this label to the list if it is not already present. Operationally, we first make a prediction with the current FSM, then benchmark that prediction against the ground-truth reference, and only *after* scoring do we use the reference to refine the dynamic state list. This ordering avoids answer leakage while allowing the system to gradually specialize the state space around emergent behaviors.

Table 12 shows that this mechanism yields a modest but consistent improvement for main characters, while leaving minor characters roughly unchanged on average. Intuitively, main characters accumulate more trajectories and thus benefit more from the incremental refinement of `other` into a richer set of recurrent emergent states. We emphasize that this dynamic mechanism is an *optional* extension rather than part of the core CFSM design. It requires online LLM intervention and access to reference trajectories, and is therefore less suitable for extremely lightweight deployments (e.g., with 1B-scale models) where our original one-shot codification is intended to operate. Nevertheless, these results

suggest a promising path for combining static, profile-derived FSMs with a thin adaptive layer that captures the long-tail of narrative-specific states encountered during extended RP.

## L  SELF-REFINEMENT DISCUSSION

Given that CFSM/CPFSM rely on LLMs to transform textual profiles into executable states and transition rules, an important complementary direction is to incorporate mechanisms that automatically assess and refine the generated logic. Recent work on iterative self-reflection demonstrates that LLMs can reliably improve their own outputs when placed in structured feedback loops. For example, Self-Refine uses model-generated critiques to progressively enhance an initial draft (Madaan et al., 2023), and Reflexion shows that explicit self-evaluation can guide agents toward more accurate reasoning trajectories (Shinn et al., 2023). In code-centric settings, self-debugging methods similarly prompt models to inspect and revise generated programs using execution feedback (Chen et al., 2024b; Zhong et al., 2024).

These developments suggest a natural extension for CFSM/CPFSM: after producing an initial FSM, the system could engage in a light self-reflection cycle by probing transitions with synthetic queries, checking reachability, or comparing behaviors against profile descriptions, and then prompting the LLM to refine the codified rules accordingly. Such self-refinement would complement our current automatic codification pipeline and offer an additional layer of reliability, particularly when scaling to richer or more ambiguous character profiles. We view this integration as a promising avenue for future work.

## M  FANDOM BENCHMARK DETAILS

Existing role-playing evaluations emphasize turn-by-turn dialogue rather than grounded, situational actions, and frequently rely on LLM-synthesized or sparsely annotated data (Wang et al., 2024b; 2025c; 2023; Chen et al., 2024a). The *Fandom Benchmark* (Peng & Shang, 2025) addresses this by framing evaluation around behavior in concrete story contexts.

Narrative segments are sourced from Fandom[3] , where `gpt-4.1` extracts character actions from each storyline segment; the preceding text becomes the *scene*. Each scene–action pair is augmented with a guiding question (also produced by `gpt-4.1`) that keeps responses reference-relevant without revealing answers. At evaluation time, the role-playing model receives a spoiler-free character profile (cleaned by `gpt-4.1`), the scene, and the question, and must predict the next action. Predictions are scored with an LLM-based NLI judge: entailed = 100, neutral = 50, contradiction = 0.

The benchmark spans six widely known artifacts across media and genres: *Suzumiya Haruhi (Haruhi)*, *K-On!*, *JOJO's Bizarre Adventure (Season 3)*, *Fullmetal Alchemist (2009)*, *A Game of Thrones (Seasons 1–3)*, and *Avatar: The Last Airbender (Book One)*: covering 83 characters and 5,141 scenes. Profiles are long-form and structured (roughly on the order of ∼1k words and ∼15 paragraphs for mains), enabling evaluation of consistent, context-aware behavior over extended narratives. Table 13 lists character summaries and example profiles. Table 14 further lists story summaries to complement background contexts.

---

[3]https://www.fandom.com/

| Series | Type | Character | Description |
|---|---|---|---|
| Haruhi | main | Haruhi | An eccentric, high-energy high schooler whose curiosity and offbeat worldview set the series' extraordinary events in motion. |
| | | Kyon | A dry-witted, pragmatic student who narrates the story and serves as Haruhi Suzumiya's reluctant but grounded companion. |
| | | Nagato | A quiet, inscrutable SOS Brigade member distinguished by her exceptional intellect and enigmatic, otherworldly origins. |
| | | Koizumi | A perpetually smiling transfer student and esper who supports the Brigade while keeping crucial secrets close. |
| | | Asahina | A shy, gentle upperclassman drafted into the SOS Brigade as their adorable, mysterious "mascot," often swept up in their schemes. |
| K-On! | main | Yui | The cheerful, airheaded lead guitarist of the light music club, whose boundless enthusiasm—and love of sweets—propels the band forward. |
| | | Ritsu | The high-spirited, mischievous drummer whose playful antics and leadership keep the group lively and in sync. |
| | | Mio | A bashful yet highly skilled bassist, gentle by nature and gifted with a keen musical sensibility. |
| | | Mugi | Tsumugi Kotobuki, a warmhearted, well-to-do keyboardist who delights in treating friends and enriching club life. |
| | | Azusa | A diligent, talented junior guitarist who quickly becomes indispensable to the club's sound and discipline. |
| | minor | Sawako | The light music club's advisor and a former hard-rocker—an encouraging teacher whose hidden metal side appears when music calls. |
| | | Nodoka | A dependable, bookish student council member and longtime friend of Yui, often helping the club with practical matters. |
| | | Ui | Yui's caring, remarkably capable younger sister, unfailingly supportive of both her sibling and the band. |
| | | Jun | A laid-back yet steadfast friend whose easygoing nature rounds out the group's everyday antics. |
| Fullmetal Alchemist | main | Edward | A brilliant, strong-willed young alchemist who sets out to restore his and his brother's bodies after a disastrous transmutation. |
| | | Alphonse | A gentle, kindhearted boy whose soul resides in a towering suit of armor, journeying with his brother to reclaim their lost selves. |
| | | Winry | A gifted automail engineer and childhood friend of the Elrics, known for her mechanical expertise and compassionate resolve. |
| | | Roy | A charismatic, ambitious State Alchemist master of flame, determined to reform the military from within. |
| | | Ling | A charming, driven prince from Xing seeking immortality while bearing a deep responsibility to his people. |
| | minor | Envy | A spiteful, shape-shifting Homunculus whose malice toward humanity fuels their cunning cruelty. |
| | | Izumi | Izumi Curtis, a fearsome alchemist and martial artist, mentors the Elrics with tough love and uncompromising standards. |
| | | Lust | A calculating Homunculus with lethal, extendable claws, pursuing enigmatic goals behind a veneer of poise. |
| | | Scar | A grim avenger marked by war, targeting State Alchemists as retribution for the devastation of his people. |
| | | Greed | A worldly, silver-tongued Homunculus driven by boundless desire, yet capable of fierce loyalty and independence. |
| | | Riza | Riza Hawkeye, an ace sharpshooter and Roy Mustang's steadfast lieutenant, defined by discipline, duty, and quiet resolve. |
| | | King Bradley | The formidable ruler of Amestris—secretly the Homunculus Wrath—who conceals deadly prowess behind a statesman's mask. |
| JOJO's Bizarre Adventure | main | Jotaro | A stoic, unflappable high schooler and "Stardust Crusaders" protagonist, famed for Star Platinum and unyielding resolve. |
| | | Polnareff | A brave, flamboyant French swordsman who joins the Crusaders, wielding the swift Stand Silver Chariot. |
| | | Joseph | A quick-thinking, larger-than-life Joestar whose trickery and bravado—"Your next line is. . ."—turn battles on their head. |
| | | DIO | A charismatic, merciless vampire whose towering ambition and "Za Warudo!" define him as an iconic nemesis. |
| | | Kakyoin | A cool, analytical ally in "Stardust Crusaders," fighting with the emerald-firing Stand Hierophant Green. |
| | | Avdol | A wise, loyal Egyptian Stand user whose Magician's Red brings fierce flame and unwavering support. |
| | | Iggy | A surly Boston Terrier Stand user with a taste for coffee gum, whose reluctant heroics prove invaluable. |
| | minor | Hol Horse | A swaggering gunslinger who manipulates foes with Emperor, a sentient revolver Stand. |
| | | Alessi | A craven antagonist whose Stand, Sethan, regressively turns victims into younger versions of themselves. |
| | | D'Arby | A devious gambler who stakes souls on deadly games, twisting chance to his favor. |
| | | Steely Dan | A sadistic schemer whose Stand, Lovers, infiltrates minds to control and torment opponents from within. |
| | | Vanilla Ice | DIO's fanatically loyal executioner, wielding Cream to erase whatever it devours from existence. |
| | | Enya | The malevolent hag of "Justice," a fog-wielding Stand user devoted to DIO's cause. |
| | | Oingo | A prank-minded shapeshifter whose Stand, Khnum, lets him mimic faces—often to comedic effect with brother Boingo. |
| | | Pet Shop | DIO's ruthless falcon sentinel, a hyper-intelligent guardian armed with the ice-forging Stand Horus. |
| | | Boingo | A timid eccentric guided by Tohth, a prophetic comic-book Stand whose literal predictions lead to bizarre outcomes. |
| A Game of Thrones | main | Tyrion | The sharp-tongued youngest Lannister, Tyrion survives Westerosi politics with wit, nerve, and gallows humor despite lifelong scorn for his stature. |
| | | Daenerys | The exiled Targaryen princess who begins as a timid pawn and grows into a resolute, power-seeking leader. |
| | | Cersei | An ambitious, calculating queen whose beauty masks ruthless devotion to her family and hold on power. |
| | | Jaime | The famed Kingslayer—charismatic, lethal, and conflicted—whose oathbound life and loyalties are anything but simple. |
| | | Robb | The honorable heir of Winterfell, thrust early into command and duty by his family's fate. |
| | | Eddard | The steadfast Lord of Winterfell, a man of austere honor serving as Warden of the North. |
| | | Arya | A fiercely independent Stark daughter who rejects courtly expectations in favor of freedom and steel. |
| | | Catelyn | The resolute Lady of Winterfell, guided by fierce maternal loyalty and hard practicality. |
| | | Sansa | The elder Stark daughter, prized for grace and courtesy, whose romantic ideals meet harsh reality. |
| | | Jon | Eddard's brooding bastard, raised at Winterfell and driven by identity, duty, and quiet resolve. |
| | | Bran | A curious seven-year-old Stark whose catastrophic fall sends him on an unexpected, fateful path. |
| | minor | Tywin | The implacable Lannister patriarch, renowned for relentless strategy, cold efficiency, and ironclad authority. |
| | | Varys | "The Spider," a soft-spoken master of whisperers whose web of informants spans the realm. |
| | | Joffrey | The cruel, petulant crown prince, vain and vindictive beneath a golden veneer. |
| | | Theon | The cocky Greyjoy heir, a Stark ward whose swagger masks divided loyalties. |
| | | Stannis | Robert's stern younger brother, grimly committed to law, claim, and uncompromising duty. |
| | | Littlefinger | Petyr Baelish, a silver-tongued master of coin who climbs through manipulation and patient plots. |
| | | Melisandre | A red priestess of Asshai, wielding unsettling faith and shadowed magic in service of R'hllor. |
| | | Jorah | A disgraced knight of Bear Island who becomes Daenerys's seasoned, steadfast adviser in exile. |
| | | Sandor | "The Hound," a brutally frank warrior scarred in flesh and spirit, sworn to guard the royal heir. |
| | | Shae | A sharp, alluring camp follower who becomes Tyrion's intimate companion amid war and intrigue. |
| | | Margaery | A poised, politically astute Tyrell whose charm and calculation flourish at court. |
| | | Davos | The plain-spoken "Onion Knight," a former smuggler turned loyal counselor to Stannis Baratheon. |
| | | Renly | Robert's magnetic youngest brother—handsome, affable, and ambitiously beloved. |
| | | Bronn | A sardonic sellsword: deadly, pragmatic, and always open to the best offer. |
| | | Brienne | The towering Lady of Tarth, unwavering in honor and skill, defying custom in pursuit of true knighthood. |
| | | Barristan | Barristan the Bold, an aging yet legendary Kingsguard knight defined by duty and dignity. |
| | | Mance | The charismatic King-Beyond-the-Wall who forges the wildlings into a single cause. |
| | | Craster | A cruel wildling patriarch infamous for incest and sacrificing sons to the Others. |
| | | Olenna | The razor-witted "Queen of Thorns," Tyrell matriarch and master of barbed politicking. |
| Avatar: The Last Airbender | main | Aang | The last Airbender and reluctant Avatar, playful by nature yet burdened to restore balance to a world at war. |
| | | Katara | A compassionate, driven waterbender from the Southern Tribe who anchors the group and challenges injustice. |
| | | Sokka | A quip-happy, inventive warrior whose boomerang and ingenuity often save the day. |
| | | Zuko | The exiled Fire Nation prince, defined by a searing quest for honor that becomes a journey of self-redefinition. |
| | minor | Iroh | A tea-loving retired general whose quiet wisdom and kindness guide his nephew toward balance. |
| | | Zhao | An ambitious Fire Nation admiral, ruthless in pursuit of glory and ruthless toward his foes. |
| | | Jet | A magnetic teen rebel leader whose anti-Fire Nation zeal blurs lines between justice and vengeance. |
| | | Yue | The gentle Northern princess whose fate entwines with the Moon Spirit and her people's survival. |
| | | Suki | The formidable Kyoshi Warrior captain, disciplined, courageous, and fiercely principled. |
| | | Appa | Aang's steadfast flying bison—gentle, mighty, and indispensable in sky and struggle alike. |
| | | Pakku | A venerable Northern master waterbender, strict in tradition yet capable of change. |

Table 13: Simple background information of characters in our experiments.

| Artifact | Concise Abstract |
|---|---|
| Haruhi Suzumiya | A high-energy, reality-bending school comedy–mystery following the eccentric Haruhi and the SOS Brigade. The narrative blends slice-of-life antics with supernatural disruptions that reflect Haruhi's unconscious influence on the world. Kyon's grounded perspective anchors the group's chaotic investigations. |
| K-On! | A warm, music-driven coming-of-age comedy centered on the Light Music Club's daily life. The story emphasizes friendship, lighthearted humor, and the gentle growth of its members as they learn instruments, perform, and bond through shared routine. |
| Fullmetal Alchemist | A dark adventure about two brothers using alchemy to restore their bodies after a forbidden transmutation. The narrative mixes political intrigue, moral conflict, and philosophical questions about sacrifice, identity, and the cost of power. |
| JOJO's Bizarre Adventure (Part 3) | A globe-trotting supernatural battle saga where Jotaro and allies confront DIO through Stand-based combat. Known for stylish fights, tactical mind games, and eccentric character dynamics that mix comedy with high-stakes drama. |
| A Game of Thrones | A sprawling political fantasy depicting rival houses struggling for power amid betrayal, war, and shifting alliances. The story juxtaposes personal ambition with looming supernatural threats, focusing on moral ambiguity and the fragility of order in Westeros. |
| Avatar: The Last Airbender | A character-driven epic about Aang's journey to master the elements and end a century-long war. Balancing humor, action, and emotional growth, the story explores identity, redemption, and the weight of responsibility. |

Table 14: Concise story information of artifacts in our experiments.

