# OpenReview forum: "Codified Finite-state Machines for Role-playing"
_ICLR.cc/2026/Conference — ICLR 2026 Poster_

### Official Review · Reviewer_VmPa · 2025-10-21

**Soundness:** 3
**Presentation:** 3
**Contribution:** 2
**Rating:** 6
**Confidence:** 4

**Summary:**

The paper introduces Codified Finite-State Machines (CFSMs), a framework that leverages large language models (LLMs) to automatically extract character states and generate executable state transition logic for role-playing (RP) agents, aiming to improve consistency and interpretability relative to prompt-based approaches. An extension, Codified Probabilistic FSMs (CPFSMs), models character states probabilistically, supporting nuanced transitions. Empirical validation includes synthetic (game-based) and real-world (Fandom Benchmark) RP tasks, demonstrating improvements in behavioral consistency, efficiency, and interpretability over established baselines.

**Strengths:**

1. Interpretability: The framework brings interpretability to state modeling in RP with executable, codified transitions derived directly from character profiles.
2. Probabilistic Extension: The CPFSM mechanism elegantly integrates stochasticity into state transitions, explicitly modeling uncertainty in RP.
3. Efficiency: CFSM delivers both accuracy and efficiency, as highlighted in Table 5.

**Weaknesses:**

1. Evaluation Scope (Generality): Empirical testing relies primarily on the Fandom Benchmark and three synthetic state machines. The real-world scenarios are derived from highly narrativized, structured data (Fandom plots) with limited diversity of state-space complexity and ambiguity. GPT-4.1 is both judge and model in several settings, and open-ended role-play evaluations rely heavily on LLM judgment. There is insufficient third-party or human evaluation of RP quality, which may limit claims of generality.
2. Limited Handling of Dynamic or Emergent States: The model assumes a fixed state set per episode. The limitations of this assumption are acknowledged in Appendix B but not addressed experimentally. Open-world RP often demands dynamic state growth or on-the-fly trait acquisition, which is not modeled or empirically probed in the present study.

**Questions:**

1. What is the meaning of "multimodal" in Line 053, and "multi-modal" in Lines 070, 080, and 092?
2. How would CFSM/CPFSM scale to open-world/large-scale RP where thousands of (possibly compositional) states, or dynamically constructed state sets, are needed? Any memory, efficiency, or codification tests on "harder" synthetic FSMs or real-world systems?

---

> ### Author Response · Authors · 2025-11-18
> **Author's Response (Part 1): Handling of Dynamic or Emergent States**
>
> We are grateful for your insightful and beneficial review! We appreciate your recognition of CFSM/CPFSM’s interpretability, probabilistic modeling, and efficiency advantages, as well as your constructive feedback on evaluation scope and dynamic state handling. We have carefully revised the paper to broaden the evaluation scope, discuss handling emergent states, and correct terminology miswriting (replacing “multimodal” with “multinomial-distribution”).
>
> ---
>
> ### **Author's Response (Part 1): Handling of Dynamic or Emergent States**
>
> Thank you for raising the question regarding CFSM’s handling of emergent or dynamically evolving states. To directly address this concern, we have proposed a new potential mechanism to support **Dynamic State Maintenance** in **New Appendix K**, an optional extension that supplements the static, profile-derived state set with lightweight refinement during long-form RP.
>
> While CFSM assumes a fixed finite set of states, real-world role-play often introduces unexpected, fine-grained states (for example, nuanced emotional combinations or situational role shifts). In the base system, such cases are mapped to a generic `other` state, which preserves correctness but loses detail. To examine how CFSM could better track this long tail of emergent behaviors, we introduce a mechanism that incrementally specializes `other` by maintaining a dynamically updated list of refined substates.
>
> When a transition falls into `other`, the LLM inspects the dialogue trajectory and proposes a more specific label. Importantly, specialization occurs only after evaluation, preventing answer leakage. Over time, the system accumulates a small, adaptive collection of emergent states closely aligned with observed behaviors.
>
> **TLDR Explanation: The CFSM now maintains a list of `other` states based on the observed history. Whenever the machine reaches `other`, it tries to select a previous state inside the list as the next state prediction.**
>
> | Artifact | Haruhi | K-On! | JOJO | FMA | AGOT | ATLA | Average |
> |----------|--------|--------|--------|--------|--------|--------|----------|
> | **Main Characters** |
> | Codified FSM             | 83.88 | 80.45 | 78.31 | 83.89 | 86.11 | 83.28 | 82.65 |
> | + Dynamic States         | 84.83 | 81.78 | 78.02 | 84.16 | 85.79 | 83.64 | 83.04 |
> | **Minor Characters** |
> | Codified FSM             | N/A   | 83.26 | 80.92 | 85.74 | 89.16 | 83.94 | 84.60 |
> | + Dynamic States         | N/A   | 82.97 | 80.27 | 85.36 | 88.79 | 85.02 | 84.48 |
> ***(New Table 12: Dynamic State Maintenance)***
>
> We observe consistent gains for main characters, who appear more frequently and therefore benefit more from emergent-state refinement. Minor characters show minimal change, likely because their behavior is already well-covered by the original profile-derived states.
>
> We emphasize that this dynamic mechanism is not part of the core CFSM pipeline. It requires online LLM intervention and access to reference trajectories **(history states)**, making it less suitable for lightweight deployments where CFSM’s statically codified structure is designed to operate. Nonetheless, these findings illustrate a promising direction for future extensions toward richer emergent-state modeling.

---

> > ### Comment · Reviewer_VmPa · 2025-11-25
> >
> > Thank you for the detailed response to my concerns.
> >
> > What is the specific online LLM used in Dynamic State Maintenance?

---

> > > ### Author Response · Authors · 2025-11-25
> > > **Re: Online LLMs used in Dynamic State Maintenance**
> > >
> > > Thanks for asking!
> > >
> > > The role assignment is consistent with the original framework: When Dynamic State Maintenance triggers "adding a new state to the other lists" (because this is a never-seen other state), the larger `gpt-4.1` is used for generation to guarantee high precision. For discriminative tasks (i.e., selecting a state from the other list), `gpt-4.1-mini` is used for choosing, just as it is used to discriminate the answers to state-transition questions in CFSM as "yes/no/unknown". We will add the details of the role assignment for Dynamic State Maintenance to the paper to improve clarity. Thank you!
> > >
> > > Please let us know if you have further questions; we are glad to answer any of them!

---

> > > > ### Comment · Reviewer_VmPa · 2025-11-25
> > > >
> > > > I understand. Thank you for your explanation. I will keep my positive score.

---

> ### Author Response · Authors · 2025-11-18
> **Author's Response (Part 2): Human Evaluation & Objective Validation**
>
> ### **Author's Response (Part 2): Human Evaluation & Objective Validation**
>
> Thank you for highlighting the importance of human validation and model-independent assessment. In the revised version, we have expanded our evaluation to include both human judgments, non-LLM-based metrics, and human revalidation to confirm the reliability of our current results.
>
> First, in the new **Social Interaction Simulation** experiment **(New Table 11 in New Appendix I)**, we conduct a small-scale human evaluation (~1k rounds of LLM role-playing as social roles for each method) on the generated multi-turn dialogues. Human annotators assessed whether each agent’s responses correctly followed its professional role manual (e.g., customer support, medical triage). Human evaluation shows strong agreement with our automated scoring, confirming that CFSM produces behaviorally coherent and contextually appropriate interactions.
>
> | Method                         | Textual Profile | Text + CoT | Text + CoT + In-Context States | CFSM      |
> |-------------------------------|-----------------|------------|--------------------------------|-----------|
> | Correct Interaction (%)        | 88.72           | 92.36      | 95.75                          | **96.13** |
> | Extra Forwarding per Round     | -               | 73.84      | 35.13                          | **1.00**  |
> ***(New Table 11: Social Interaction Simulation Results)***
>
> Second, we included a model-independent metric, **ROUGE-L** (used in previous literature [1]), alongside the NLI-based consistency score. We place the ROUGE-L performance of CFSM together with key baselines **(Some are new baselines in response to Reviewer u1Hi's suggestion for broader comparison)** in **New Table 10**. ROUGE-L evaluation also shows that CFSM consistently achieves the highest performance. This alignment between semantic and surface-level measures further supports the robustness of our evaluation.
>
> | Artifact / Method | Haruhi | K-On! | JOJO | FMA | AGOT | ATLA | Average |
> |--------------------|--------|-------|------|-----|------|------|---------|
> | **Main Characters** ||||||||
> | Codified Profile | 20.57 | 18.50 | 18.82 | 18.82 | 20.81 | 18.95 | 19.41 |
> | PromptTrans | 20.09 | 18.09 | 18.72 | 18.91 | 20.60 | 18.68 | 19.18 |
> | Character Updating | 20.71 | 18.05 | 18.53 | 18.58 | 20.10 | 18.81 | 19.13 |
> | Plot Summary | 21.11 | 18.22 | 17.98 | 18.93 | 20.52 | 18.51 | 19.21 |
> | Codified FSM | 20.61 | 19.09 | 18.78 | 19.32 | 21.28 | 19.35 | **19.74** |
> | **Minor Characters** ||||||||
> | Codified Profile | — | 21.21 | 18.09 | 21.40 | 21.66 | 19.17 | 20.31 |
> | PromptTrans | — | 20.18 | 18.13 | 20.78 | 21.27 | 19.06 | 19.88 |
> | Character Updating | — | 19.51 | 18.39 | 21.61 | 20.81 | 18.87 | 19.84 |
> | Plot Summary | — | 20.19 | 18.38 | 21.16 | 21.67 | 18.93 | 20.07 |
> | Codified FSM | — | 20.93 | 18.74 | 22.88 | 21.49 | 20.02 | **20.81** |
> ***(New Table 10: Main RP Performance Comparison (ROUGE-L Score))***
>
> Finally, to ensure the validity of the NLI metric itself, we performed an additional manual verification of 600 samples (200 each labeled as `entailed`, `neutral`, and `contradicted`). Human judgments matched the NLI predictions with $91.0%$, $89.5%$, and $88.0%$ accuracy, respectively, which is consistent with earlier findings from prior work, as NLI is a rather basic NLP ability mastered by LLMs. These results confirm that our NLI-based scoring provides a reliable proxy for human evaluation of character consistency. We have correspondingly updated the revalidation results to **Section 5.1**.
>
> - [1] CoSER: Coordinating LLM-Based Persona Simulation of Established Roles, ICML 2025

---

> ### Author Response · Authors · 2025-11-18
> **Author's Response (Part 3): Clarification on Terminology Miswriting**
>
> ### **Author's Response (Part 3): Clarification on Terminology Miswriting**
>
> We thank the reviewer for catching the confusion around the term **“multimodal”** appearing in several early sections. This was an unintended misspelling, and the correct term should be **“multinomial”**, referring to the multinomial-distribution-based transitions in CPFSM that model multiple possible next states with probabilistic weights.
>
> We have corrected all occurrences of “multimodal” (and “multi-modal”) to “multinomial” throughout the revised version and updated the surrounding explanation to clarify that CPFSM models stochastic, weighted transitions among character states, rather than multi-sensory or multi-input modalities.
>
> ---
>
> We truly appreciate your thoughtful review and valuable insights, which helped us refine both the technical framing and clarity of our work. If you have any additional questions or suggestions during the discussion phase, please feel free to raise them. We will remain fully engaged throughout the author response period and will address every point promptly and respectfully to ensure our revisions meet your expectations.

---

> ### Author Response · Authors · 2025-11-18
> **Response Appendix: Details of Conversational Social Simulation Tests**
>
> In response to Reviewer u1Hi's suggestion, we added a new experiment titled **"Social Interaction Simulation"** in **New Appendix I** of the revised version.
>
> In this experiment, we construct five professional roles that require strict adherence to behavioral rules:
> *Customer Support Agent, Medical Triage Nurse, Teacher or Tutor, Interviewer, and Project Manager.*
> For each role, we create ten diverse user personas with different communication styles and goals. Each conversation proceeds as a goal-driven multi-turn interaction with a maximum of 20 rounds. Dialogues terminate early when the task is completed. We then manually evaluate whether each agent’s response correctly follows its role manual.
>
> ---
>
> **Example of role manual (Beginning Part):**
> ```
> Customer Support Agent — Behavior Manual
>
> The customer support agent begins by welcoming the user and clarifying the issue.
> The agent’s first task is to restate the problem in simple terms and check basic constraints such as account status, product type, or affected feature.
>
> If the problem is unclear, the agent continues asking focused questions until the details are sufficient. A cycle of clarifying questions may continue as long as the user’s descriptions remain vague, contradictory, or incomplete...
> ```
>
> ---
>
> **Example of tester's persona**
> ```
> A polite elderly user who is unfamiliar with modern interfaces and often misinterprets technical terminology. They remain calm but frequently express uncertainty about each step.
> ```
>
> ---
>
> **Comparison Settings**
>
> We compare four settings:
>
> - Textual Profile: Direct response generation based only on the role description (manual).
>
> - Textual Profile + CoT: The model performs reasoning before responding.
>
> - Textual Profile + CoT + In-context States: CoT-based state inference plus explicit recording of previous transitions in the prompt to reduce re-derivation.
>
> - CFSM: Uses the codified transition function to update states without requiring CoT.
>
> ---
>
> **Results**
>
> | Method                         | Textual Profile | Text + CoT | Text + CoT + In-Context States | CFSM      |
> |-------------------------------|-----------------|------------|--------------------------------|-----------|
> | Correct Interaction (%)        | 88.72           | 92.36      | 95.75                          | **96.13** |
> | Extra Forwarding per Round     | -               | 73.84      | 35.13                          | **1.00**  |
> ***(New Table 11: Social Interaction Simulation Results)***
>
> The results, reproduced below from **New Table 11** in the revised version, confirm a similar pattern found in our synthetic validation experiments: CFSM achieves the highest behavioral correctness and dramatically reduces overhead caused by repeated reasoning. A common observed issue in **Textual Profile** and **Text + CoT** is their limit in transitioning back to a previous state following the manual ($s_1 \rightarrow s_2 \rightarrow s_1$) because it tends to copy its latest conversation way or brutally proceed forward without carefully checking for its correct next state.

---

### Official Review · Reviewer_wsJe · 2025-10-31

**Soundness:** 3
**Presentation:** 4
**Contribution:** 3
**Rating:** 6
**Confidence:** 4

**Summary:**

This paper proposes CFSM and CPFSM for roleplaying tasks and games with LLMs to ensure behavioral coherence. This paper contributes by offering a framework that defines character transitions and extends it to a probabilistic version to offer multi-modal transitions. Authors have compared and presented the limitations of the LLMs in Section 4 while also showing that CFSM is superior to the baselines compared. The Authors have also discussed the setup and case studies considered for the real plot experiment. The number of models used for the experiment is good, considering the innovation is a technical advancement with LLM rather than the performance of the LLM itself.

**Strengths:**

1) The described methods work on various artifacts mentioned in the results, while
demonstrating the strong performance against the baselines.
2) The paper mentions the computational complexity for the both methods and shows
faster and efficient codification for the proposed methods.
3) This paper includes a very detailed analysis section mentioning synthetic and real plot
experiments, and is tested with multiple LLM models and techniques, and has various
kind of plots and scenes from various genres.

**Weaknesses:**

1) The “preliminary and denotation” introduces the necessary terminology but lacks
examples and a lucid explanation, which can be really helpful for the readers and the
general audience unaware of such methods.
2) The multi-modality and reactions of CPFSM lack depth and can be explained more
clearly.
3) The real plot experience can briefly explain one of the artifacts used in the work as a
running example. Not having this makes it lless intuitive for new readers.

**Questions:**

Detailed suggestions:
1) NLI full form could be better when referenced first.
2) Best@K can be explained.
3) Figure 4 Caption: There is no space between CFSM&amp;CPFSM.
4) Line 109 - evolve should be “evolves”.
5) Line 223 - w_i,j “is” then normalized.
6) Line 225 - in binary_questions, how the logits w_i,j are derived from the
“Yes/no/unknown” question. This can be explained in detail.
7) In Tables 2, 3, and 4, 6, the units of measurement are missing; it would be reader-
friendly to add them.
8) In the baseline, #Character is mentioned for each show, but lacks a reference to it. For
example, Haruhi mentions 5 Characters and AGOT 11, but a brief description of at least
one of the artifacts, such as JOJO, its characters, and the context of the scene, and
profiles would be intuitive for readers to analyze the results better.
9) Multi-modality and transitions of CPFSM can be explained in detail.

---

> ### Author Response · Authors · 2025-11-17
> **Author's Response: Writing Clarity and Better Presentation**
>
> ### **Author's Response: Writing Clarity and Better Presentation**
>
> We sincerely thank you for your positive evaluation and detailed feedback on our work! We have carefully revised the paper to address all writing and presentation concerns you raised. In detail:
>
> - We added illustrative examples using ***Jotaro Kujo***'s state transition case from ***JOJO's Bizarre Adventure*** in the “Preliminary and Denotation” section to clarify key terminology and improve readability for general audiences.
>
> - We included two data samples (from ***Haruhi*** and ***AGOT***) in the real plot experiment section **(New Table 2)** to make the benchmark clearer to readers without context knowledge.
>
> - The CPFSM explanation has been expanded to better describe its **multinomial transition behavior** (previously miswritten as **“multimodal”**), and the text now explicitly states **“multinomial-distribution transitions.”** where previously mentioned **multi-modality**.
>
> - We mention NLI score as the default reported metric in **New Table 3**, and **Figure 4** formatting has been corrected.
>
> - We also added the full form of NLI (Natural Language Inference), an explanation of Best@K, and clarified how the binary question logits are derived (the log probability of "True" from the NLI classifier).
>
> - Minor editorial improvements have been made: spacing, grammar (e.g., “evolve” → “evolves”), and normalization notation fixes.
>
> We appreciate your constructive suggestions; they helped improve the paper’s clarity and accessibility for a broader audience. If you have any follow-up concerns about our presentation, please don't hesitate to raise them during the author response period. We will stay actively involved during the whole discussion and provide prompt responses to you with the highest respect for your efforts in improving our work!

---

> ### Author Response · Authors · 2025-11-27
> **Invitation to further discussion on CFSM**
>
> Dear Reviewer wsJe,
>
> We sincerely respect your efforts and insights devoted to reviewing our submission! We will also keep our "responsible interaction" promise in our previous response by staying active to address any potential further concerns from you during the discussion period.
>
> We completely understand and value your time, which is precious for every researcher. But since it's only 1 week left before the end of the discussion period, we would like to kindly invite you to further discuss whether your concerns have been addressed. This will also give us enough time for potential further content (e.g., experiment/writing) to make sure the discussion results in high-quality refinement to the work.
>
> Thanks again for your contribution, and we eagerly look forward to further discussion with you!

---

### Official Review · Reviewer_u1Hi · 2025-11-01

**Soundness:** 3
**Presentation:** 3
**Contribution:** 3
**Rating:** 4
**Confidence:** 3

**Summary:**

This paper proposes Codified Finite-State Machines (CFSM), a framework that enhances character consistency in LLM role-playing by automatically extracting and codifying character states from textual profiles. CFSMs transform character descriptions into explicit FSM structures using LLM-generated logic, grounding behavior in interpretable state transitions. A probabilistic extension, CPFSM, further models transition uncertainty by maintaining distributions over states. The paper evaluates the approach in both synthetic state modeling tasks and large-scale narrative role-play (via the Fandom Benchmark) and demonstrates improved consistency, interpretability, and transition traceability compared to prompt-based methods. Ablation and cost analyses show that CFSM/CPFSM are scalable and effective, offering a hybrid symbolic–neural approach to stateful role-play generation.

**Strengths:**

- The codification of character logic via FSMs, driven by LLMs, presents a novel mechanism to preserve behavioral coherence in long-form role-playing.

- Experimental results show a clear improvement in behavioral consistency after introducing CFSM. Whether in synthetic tasks (e.g., Mario state transitions) or real narrative scenarios, characters’ state transitions become more coherent and believable. CFSM and CPFSM effectively reduce the confusion and inconsistency commonly observed in prompt-based methods. Notably, CPFSM enhances the subtlety and realism of character responses by modeling weighted reactions across multiple plausible actions through probabilistic transitions.

- Unlike prompt-only state modeling, CFSMs generate explicit transition rules, enabling better control and debuggability in interactive settings.

**Weaknesses:**

The proposed framework heavily depends on the LLM to extract states and generate transition rules. If the LLM-produced code contains errors or omissions, it may compromise the correctness of the resulting finite-state machine. The paper provides limited discussion on how to validate or correct the logic generated by the LLM, leaving the reliability of the approach partially contingent on the quality of the LLM’s rule extraction process.

Another concern lies in the current evaluation, which primarily focuses on the Synthetic Validation setup and the Fandom Benchmark — both emphasizing narrative-driven scenarios and character-centric tasks. While these datasets are structurally sound and semantically rich, it would strengthen the work to include more conventional evaluation settings, such as open-domain human–AI dialogue, task-oriented dialogue systems, or social simulation environments. Extending the experiments to broader multi-turn dialogue contexts would better demonstrate the generality and transferability of the proposed CFSM/CPFSM framework.

In addition, incorporating more objective and independent evaluation metrics would provide a more comprehensive assessment of model performance. The selection of baselines also appears somewhat limited: although Codified Profile and PromptTrans offer partial validation of the proposed design, the absence of stronger or more up-to-date baselines weakens the comparative significance. Including results against more advanced or representative methods could substantially enhance the paper’s empirical rigor and impact.

**Questions:**

See in weakness.

---

> ### Author Response · Authors · 2025-11-17
> **Author's Response (Part 1): Conversational Social Simulation Tests**
>
> Thank you for your thoughtful and constructive review! We appreciate your recognition of the novelty and impact of Codified Finite-State Machines (CFSM) for improving state-consistent role-playing. We also value your detailed suggestions on experimental scope and result validation. In the revised version, we have incorporated additional conversational social-simulation experiments, added model-agnostic evaluation metrics, expanded baseline comparisons, and clarified how to mitigate potential codification errors in CFSMs. We hope these updates effectively address your concerns.
>
> ---
>
> ### **Author's Response (Part 1): Conversational Social Simulation Tests (Multi-round & Dialogue system & Social simulation)**
>
> Thank you for your suggestion to evaluate CFSM in broader dialogue and social simulation settings. We fully agree that real-world conversational environments provide an important validation of state maintenance quality. In response, we added a new experiment titled **"Social Interaction Simulation"** in **New Appendix I** of the revised version.
>
> In this experiment, we construct five professional roles that require strict adherence to behavioral rules:
> *Customer Support Agent, Medical Triage Nurse, Teacher or Tutor, Interviewer, and Project Manager.*
> For each role, we create ten diverse user personas with different communication styles and goals. Each conversation proceeds as a goal-driven multi-turn interaction with a maximum of 20 rounds. Dialogues terminate early when the task is completed. We then manually evaluate whether each agent’s response correctly follows its role manual.
>
> ---
>
> **Example of role manual (Beginning Part):**
> ```
> Customer Support Agent — Behavior Manual
>
> The customer support agent begins by welcoming the user and clarifying the issue.
> The agent’s first task is to restate the problem in simple terms and check basic constraints such as account status, product type, or affected feature.
>
> If the problem is unclear, the agent continues asking focused questions until the details are sufficient. A cycle of clarifying questions may continue as long as the user’s descriptions remain vague, contradictory, or incomplete...
> ```
>
> ---
>
> **Example of tester's persona**
> ```
> A polite elderly user who is unfamiliar with modern interfaces and often misinterprets technical terminology. They remain calm but frequently express uncertainty about each step.
> ```
>
> ---
>
> **Comparison Settings**
>
> We compare four settings:
>
> - Textual Profile: Direct response generation based only on the role description (manual).
>
> - Textual Profile + CoT: The model performs reasoning before responding.
>
> - Textual Profile + CoT + In-context States: CoT-based state inference plus explicit recording of previous transitions in the prompt to reduce re-derivation.
>
> - CFSM: Uses the codified transition function to update states without requiring CoT.
>
> ---
>
> **Results**
>
> | Method                         | Textual Profile | Text + CoT | Text + CoT + In-Context States | CFSM      |
> |-------------------------------|-----------------|------------|--------------------------------|-----------|
> | Correct Interaction (%)        | 88.72           | 92.36      | 95.75                          | **96.13** |
> | Extra Forwarding per Round     | -               | 73.84      | 35.13                          | **1.00**  |
> ***(New Table 11: Social Interaction Simulation Results)***
>
> The results, reproduced below from **New Table 11** in the revised version, confirm a similar pattern found in our synthetic validation experiments: CFSM achieves the highest behavioral correctness and dramatically reduces overhead caused by repeated reasoning. An common observed issue in **Textual Profile** and **Text + CoT** is their limit in transitioning back to a previous state following the manual ($s_1 \rightarrow s_2 \rightarrow s_1$) because it tends to copy its latest conversation way or brutally proceed forward without a carefully checking for its correct next state. This also supports your expectation that CFSM transfers well to social dialogue domains and strengthens the overall contribution of our work, thank you!

---

> > ### Author Response · Authors · 2025-11-17
> > **Author's Response (Part 2): Expanded Baselines and Model-independent Metrics**
> >
> > ### **Author's Response (Part 2): Expanded Baselines and Model-independent Metrics**
> >
> > We appreciate your suggestion to expand our baseline coverage and incorporate additional evaluation metrics that do not rely on model-generated judgments. In the revised version, we address both points.
> >
> > First, we include two recently proposed state-tracking approaches as additional baselines in **New Table 3**:
> >
> > - **Character Updating,** introduced in CharacterBox@NAACL2025 [1].
> >
> > - **Plot Summary,** proposed in CoSER@ICML2025 [2].
> >
> > Both methods employ multi-stage prompting or history consolidation for latent state inference. Compared with our existing PromptTrans baseline, these systems are more complex and use substantial prompt engineering to maintain character coherence. We also note that our Codified Profile baseline corresponds to a NeurIPS 2025 accepted work according to the conference's agenda, which represents an up-to-date baseline in the original work.
> >
> > Second, we add a **model-independent metric, ROUGE-L**, following the metric selection in previous work [2], to evaluate content similarity between expected and generated character behaviors. This ensures that our conclusions are not tied to a single scoring model. The results are reported in **New Table 10**.
> >
> > Together, these additions broaden the empirical foundation and allow for a more comprehensive comparison across methodologies.
> >
> > | Artifact / Method | Haruhi | K-On! | JOJO | FMA | AGOT | ATLA | Average |
> > |--------------------|--------|-------|------|-----|------|------|---------|
> > | **Main Characters** ||||||||
> > | Codified Profile | 82.77 | 80.49 | 76.19 | 82.58 | 84.47 | 81.66 | 81.36 |
> > | PromptTrans | 82.86 | 79.68 | 75.07 | 82.66 | 84.96 | 82.17 | 81.23 |
> > | Character Updating | 83.63 | 79.23 | 75.08 | 82.48 | 85.87 | 82.51 | 81.47 |
> > | Plot Summary | 83.19 | 79.51 | 76.12 | 82.91 | 85.48 | 83.00 | 81.70 |
> > | Codified FSM | 83.88 | 80.45 | 78.31 | 83.89 | 86.11 | 83.28 | **82.65** |
> > | **Minor Characters** ||||||||
> > | Codified Profile | — | 81.54 | 80.58 | 85.91 | 88.91 | 82.16 | 83.82 |
> > | PromptTrans | — | 82.17 | 81.72 | 83.16 | 86.88 | 82.61 | 83.31 |
> > | Character Updating | — | 82.13 | 80.39 | 85.49 | 89.13 | 82.57 | 83.59 |
> > | Plot Summary | — | 82.34 | 80.61 | 85.10 | 88.76 | 82.79 | 83.92 |
> > | Codified FSM | — | 83.26 | 80.92 | 85.74 | 89.16 | 83.94 | **84.60** |
> > ***(New Table 3: Main RP Performance Comparison (NLI Score))***
> >
> >
> > | Artifact / Method | Haruhi | K-On! | JOJO | FMA | AGOT | ATLA | Average |
> > |--------------------|--------|-------|------|-----|------|------|---------|
> > | **Main Characters** ||||||||
> > | Codified Profile | 20.57 | 18.50 | 18.82 | 18.82 | 20.81 | 18.95 | 19.41 |
> > | PromptTrans | 20.09 | 18.09 | 18.72 | 18.91 | 20.60 | 18.68 | 19.18 |
> > | Character Updating | 20.71 | 18.05 | 18.53 | 18.58 | 20.10 | 18.81 | 19.13 |
> > | Plot Summary | 21.11 | 18.22 | 17.98 | 18.93 | 20.52 | 18.51 | 19.21 |
> > | Codified FSM | 20.61 | 19.09 | 18.78 | 19.32 | 21.28 | 19.35 | **19.74** |
> > | **Minor Characters** ||||||||
> > | Codified Profile | — | 21.21 | 18.09 | 21.40 | 21.66 | 19.17 | 20.31 |
> > | PromptTrans | — | 20.18 | 18.13 | 20.78 | 21.27 | 19.06 | 19.88 |
> > | Character Updating | — | 19.51 | 18.39 | 21.61 | 20.81 | 18.87 | 19.84 |
> > | Plot Summary | — | 20.19 | 18.38 | 21.16 | 21.67 | 18.93 | 20.07 |
> > | Codified FSM | — | 20.93 | 18.74 | 22.88 | 21.49 | 20.02 | **20.81** |
> > ***(New Table 10: Main RP Performance Comparison (ROUGE-L Score))***
> >
> > Across both metrics, the more elaborate prompting-based baselines exhibit modest gains compared with simpler methods. However, CFSM consistently achieves the strongest performance. This indicates that CFSM introduces not only an alternative formulation, but a distinct methodological advantage for character state modeling that prompting alone cannot easily replicate.
> >
> > We hope these expanded baselines and model-independent evaluations address your concerns and provide stronger empirical grounding for our claims.
> >
> > - [1] CharacterBox: Evaluating the Role-Playing Capabilities of LLMs in Text-Based Virtual Worlds, NAACL 2025
> >
> > - [2] CoSER: Coordinating LLM-Based Persona Simulation of Established Roles, ICML 2025

---

> ### Author Response · Authors · 2025-11-17
> **Author's Response (Part 3): Self-refinement Discussion**
>
> ### **Author's Response (Part 3): Self-refinement Discussion**
>
> Thank you for highlighting the importance of validating the correctness of LLM-generated transition rules. We agree that adding mechanisms for automatic refinement would further strengthen the reliability of CFSM and CPFSM. In the revised version, we added a discussion in **Appendix L** on this topic.
>
> Recent advances in iterative self-reflection show that large language models can improve their own outputs through structured feedback loops. Methods such as Self-Refine [3] and Reflexion [4] demonstrate that model-generated critiques and self-evaluation can reliably enhance reasoning quality, while code-oriented approaches to self-debugging [5,6] similarly use execution feedback to iteratively correct generated programs. These findings naturally complement the CFSM pipeline.
>
> Building on this literature, we propose extending CFSM/CPFSM with a lightweight self-refinement stage. After producing an initial FSM, the system could automatically probe transitions with synthetic queries, verify reachability, or compare expected and codified behaviors against the source profile. The LLM would then revise any inconsistent or unreachable rules, adding a secondary layer of validation on top of the existing codification process.
>
> We believe this integration represents a promising direction for future work, especially as CFSM scales to more complex or ambiguously defined character profiles.
>
> - [3] Self-Refine: Iterative Refinement with Self-Feedback, NeurIPS 2023
> - [4] Reflexion: Language Agents with Verbal Reinforcement Learning, NeurIPS 2023
> - [5] Teaching Large Language Models to Self-Debug, ICLR 2024
> - [6] Debug like a Human: A Large Language Model Debugger via Verifying Runtime Execution Step-by-step, ACL Findings 2024
>
> ---
>
> We hope the responses and additional results above have addressed your concerns. We will stay active during the whole author response period to promptly answer any follow-up questions from you to express our greatest respect for your devoted efforts in improving the quality of our work!

---

> ### Author Response · Authors · 2025-11-18
> **Response Appendix: A Running Case of Social Interaction Agent driven by CFSM**
>
> The following is a summarized example to show how CFSM is driving an agent **(Customer Support Agent)** in social activity. The tester's persona is "A polite elderly user who is unfamiliar with modern interfaces and often misinterprets technical terminology. They remain calm but frequently express uncertainty about each step."
>
> ---
>
> **Turn 1**
>
> USER: Reports Wi-Fi stopped working since yesterday; unsure what to do.
>
> AGENT: Greets warmly, acknowledges frustration, asks clarifying questions about errors and other devices.
>
> STATE: Agent is greeting the user and framing the issue.
>
> ---
>
> **Turn 2**
>
> USER: Says a red X appears, clicking doesn’t do much; phone works fine.
>
> AGENT: Confirms details, expresses empathy, asks about Wi-Fi switch, restart, and OS type.
>
> STATE: Agent is clarifying the problem and gathering diagnostic details.
>
> ---
>
> **Turn 3**
>
> USER: Unsure about physical Wi-Fi switch; restarted earlier; laptop is likely Windows.
>
> AGENT: Guides through checking function key (Fn + wireless key), Wi-Fi icon, and Airplane mode.
>
> STATE: Agent is clarifying the problem and gathering diagnostic details.
>
> ---
>
> **Turn 4**
>
> USER: Tried function key, no effect; clicking shows “No connections available”; no airplane icon.
>
> AGENT: Moves to deeper troubleshooting—checking Device Manager and running Windows Network Troubleshooter.
>
> STATE: Agent is delivering a solution or step-by-step instructions.
>
> ---
>
> **Turn 5**
>
> USER: Doesn’t know how to open Device Manager; feels lost in menus.
>
> AGENT: Reassures user and gives simple step-by-step instructions for opening Device Manager.
>
> STATE: Agent is clarifying the problem and gathering diagnostic details.
>
> ---
>
> **Turn 6**
>
> USER: Successfully opened Device Manager and found “Network adapters.”
>
> AGENT: Guides user to expand Network adapters and inspect wireless adapter status.
>
> STATE: Agent is delivering a solution or step-by-step instructions.
>
> ---
>
> **Turn 7**
>
> USER: Wireless adapter shows “Disable device,” meaning it’s enabled; no down arrow on icon.
>
> AGENT: Confirms adapter is enabled; instructs user to run Windows Network Troubleshooter.
>
> STATE: Agent is delivering a solution or step-by-step instructions.
>
> ---
>
> **Turn 8**
>
> USER: Troubleshooter couldn't identify the problem.
>
> AGENT: Offers new steps: checking Wi-Fi status in Settings and restarting the adapter.
>
> STATE: Agent is clarifying the problem and gathering diagnostic details.
>
> ---
>
> **Turn 9**
>
> USER: Wi-Fi settings show “No networks found”; disabling/enabling adapter didn’t help.
>
> AGENT: Suggests a full Network Reset to reinstall adapters and restore defaults.
>
> STATE: Agent is delivering a solution or step-by-step instructions.
>
> ---
>
> **Turn 10**
>
> USER: Ran Network Reset; computer restarting; asks what to do next.
>
> AGENT: Tells user to reconnect after restart and check Wi-Fi list.
>
> STATE: Agent is delivering a solution or step-by-step instructions.
>
> ---
>
> **Turn 11**
>
> USER: After restart, networks appear; connects successfully using password.
>
> AGENT: Confirms fix; asks user to test browser to ensure internet works.
>
> STATE: Agent is delivering a solution or step-by-step instructions.
>
> ---
>
> **Turn 12**
>
> USER: Browser loads correctly; expresses gratitude for patient guidance.
>
> AGENT: Closes warmly, offers future help.
>
> STATE: Agent is not actively engaged (conversation resolved).
>
> ---
>
> We can view a smooth state back shifting to `detail gathering` in **Turn 7 $\rightarrow$ Turn 8** when the agent faces an ambiguous issue. A pure textual manual-driven agent fails to detect such nuance to propose follow-up questions to further diagnose the issue.

---

> ### Author Response · Authors · 2025-11-27
> **Invitation to further discussion on CFSM**
>
> Dear Reviewer u1Hi,
>
> We sincerely respect your efforts and insights devoted to reviewing our submission! We will also keep our "responsible interaction" promise in our previous response by staying active to address any potential further concerns from you during the discussion period.
>
> We completely understand and value your time, which is precious for every researcher. But since it's only 1 week left before the end of the discussion period, we would like to kindly invite you to further discuss whether your concerns have been addressed. This will also give us enough time for potential further content (e.g., experiment/writing) to make sure the discussion results in high-quality refinement to the work.
>
> Thanks again for your contribution, and we eagerly look forward to further discussion with you!

---

### Author Response · Authors · 2025-12-03
**Discussion Period Summary**

Dear Reviewers, AC, SAC, and PC.

We sincerely appreciate the efforts you devoted to reviewing and the discussion period. Thank you all!

For your convenience, we would like to provide a brief and unbiased summary of the background and progress of the discussion period.

---

### **What reviewers agree on? (Strength)**

**Novelty (u1Hi-S1, wsJe-Summary, VmPa-S2):** Our proposed Codified (Probabilistic) Finite-State Machine (CFSM/CPFSM) is a useful but currently untouched design for the role-playing domain, thus resulting in a novel contribution to the field.

**Performance & Efficiency (u1Hi-S2, wsJe-S1&S2, VmPa-S3):** CFSM/CPFSM extracts state transition rules from textual profiles and models them as executable functions (maintained by yes/no questions for models to answer efficiently), which guarantees the transition consistency and efficiency.

**Interpretability & Transparency (u1Hi-S3, wsJe-S3, VmPa-S1):** CFSM/CPFSM provide codes that contain explicit transition logics that are readable by humans. During the execution of CFSM/CPFSM for state modeling, the trajectory of the state (distribution) transition is accessible for sampling or intermediate checking.

---

### **What is discussed during the period? (Weakness & Question)**

**Evaluation Scope (u1Hi-W2&W3, VmPa-W2&Q2):**

**(1)** Reviewer **u1Hi** proposes that CFSM/CPFSM can be evaluated beyond the fictional context that validates its superior performance and efficiency.

**Response:** We have correspondingly added a social interactive experiment (agency, triage) to validate that CFSM/CPFSM can be extended to real-world role-playing usage.

**(2)** Reviewer **VmPa** mentions that we can have a third-party validation on the reliability of LLM-based NLI or have other model-independent metrics to validate the conclusion.

**Response:** We mention that NLI reliability for role-playing judging has been validated in previous literature [3], and add a human validation of NLI reliability in our results. We have also added the ROUGE metric as a model-independent evaluation, which supports the conclusions drawn from the NLI evaluation.

**(3)** Reviewer **u1Hi** proposes that more baselines can be added to further support the contribution of CFSM/CPFSM.

**Response**: We add 2 extra state modeling methods from recent impactful works [1, 2] to the Table and mention we have already involved a recent strong method (codified profile) [3] as one of the baselines.

**CFSM/CPFSM Add-ons (u1Hi-W1, VmPa-W1):**

**(1)** Reviewer **u1Hi** asks for potential mechanisms to address the reliance on the CFSM/CPFSM initialization, i.e., addressing potential misinterpretation in the originally codified logic.

**Response:** We discuss two scenarios for self-improvement: When ground-truth (e.g., real scene & character's action pairs) is available, we can use the difference between the machine's prediction and ground-truth for debugging. Otherwise, we can take advantage of self-reflection & debugging mechanisms like synthesizing test cases for self-improvement. As code, CFSM/CPFSM can be debugged in the same way as code.

**(2)** Reviewer **VmPa** asks for a method to handle dynamic states that appear in the storyline.

**Response:** We discuss a method to fine-granularize the other state in the current design for CFSM/CPFSM. The dynamic states are registered and retrieved in the other state. Experiments also show such a mechanism to be beneficial for the main characters in the storyline.

**Writing & Clarity (VmPa-Q1, wsJe-W&Q)**

Both reviewers pointed out that the presentation could be clearer in a few places, including the preliminaries, the explanation of CPFSM transitions, and some overloaded terminology.

**Response:** In the revised version, we (i) clarified all technical terms (for example, correcting “multimodal” to “multinomial-distribution transitions”), (ii) added small running examples in the preliminaries and real-plot section to make the setting more accessible, (iii) defined NLI and Best@K on first use and added missing units to tables, and (iv) briefly introduced key artifacts (such as JOJO and AGOT) to give readers better context for the results, while also fixing minor typos and formatting issues.

---

We hope this summary helps the AC, SAC, and PC see that the main concerns raised during the discussion have been directly and concretely addressed in the revised version. Thank our reviewers again for your time, effort, and thoughtful guidance!

- [1] CharacterBox: Evaluating the Role-Playing Capabilities of LLMs in Text-Based Virtual Worlds, **NAACL 2025**

- [2] CoSER: Coordinating LLM-Based Persona Simulation of Established Roles, **ICML 2025**

- [3] Codifying Character Logic in Role-Playing, **NeurIPS 2025**

---

### Meta-Review · Area_Chair_np84 · 2026-01-18

**Summary:**

This paper proposes to codify the FSM of a character's internal state transition based on its profile. The generated FSM is then applied during the role-playing stage to ensure consistent content.

I recommend acceptance in respect to the reviewer's comments, with 2 weak accept (6) and 1 weak reject (4). I share some concerns with reviewers and also have some unmentioned cocnerns over the content of the paper as detailed below, so I wouldn't mind if the paper gets rejected.

The major concerns on this paper comes from the following aspects:
- Dependence on the codifiablity (e.g., cannot handle emergent states) and coding quality: a strong coder model is needed for the small RP model experiments, while the gain for the large RP model is minimal even with the large coder model.
- Scope of the evaluation and Insignificant improvements:
    - Evaluations are limited to synthetic and Fandom datasets.
    - The synthetic ones are all essentially TC^0 problems that can be modeled directly by transformers (if trained on it as shown in previous work [1]), indicating that the failure is more likely to come from a lack of in-domain training for state tracking. The Fandom is still limited in the topics and have the risk of information leaking from pre-training data in predicting famous characters' behaviors. A social simulation is added, but still highly synthetic.
    - The NLI-based scorer has an human-agreement of ~90%, while the improvements on the large RP model is only within 1.5%, and the smaller model of 3% with strong subset bias (improves only on 2 subsets). Objective scores with stronger improvements would strengthen this work.

[1]  W. Merrill and A. Sabharwal. The expressive power of transformers with chain of thought.

**Reviewer Concerns:**

The authors added more recent baselines to resolve reviewer's concern on a wek baseline. They also included a new social simulation tests to expand the scope of the evaluation.

**Reviewer Scores:**

They should still be keeping their scores given more discussions.

---

### Decision · Program_Chairs · 2026-01-26

Accept (Poster)